# SOFT QUANTIZATION ACTIVATION FUNCTIONS FOR DEEP LEARNING

## ABSTRACT

Activation functions (AFs) are a cornerstone of deep learning, providing the crucial nonlinearity needed for network expressiveness. However, widely used AFs like ReLU and GELU are fixed and non-adaptive, offering limited nonlinearity and often necessitating larger, more complex architectures to capture intricate functions. This paper introduces a new family of trainable, architecture-agnostic AFs called Soft Quantization Activation Functions (SQUAFs). We show theoretically that SQUAFs can approximate any continuous nonlinear one-dimensional function with arbitrary precision. Our extensive experiments demonstrate that networks equipped with SQUAFs consistently outperform their counterparts using existing AFs across diverse tasks. Specifically, we achieve orders-of-magnitude error reduction in function fitting, up to 25.27 dB gain in image fitting, and significant accuracy improvements in image classification and large language model (LLM) fine-tuning. Moreover, SQUAFs (1) enable smaller models to surpass larger ones trained with conventional AFs, and (2) can reduce the inter-device communication cost in model-parallel settings by up to 9-fold while still improving accuracy. These results highlight SQUAFs as a simple yet powerful drop-in replacement for standard AFs, offering both theoretical expressiveness and practical performance gains.

## 1 INTRODUCTION

The remarkable success of deep neural networks (DNNs) across computer vision, natural language processing, and other domains is largely attributed to their ability to approximate highly nonlinear relationships within data. This nonlinear capability arises from several sources, such as softmax in attention (Vaswani, 2017) and classifiers, max-pooling, batch normalization (training mode) (Ioffe & Szegedy, 2015), layer normalization (Ba et al., 2016), and quantization (Yang & Hamidi, 2025; Salamah et al., 2025), but is fundamentally provided by activation functions (AFs). Positioned between the layers of a network, AFs are the primary mechanism through which nonlinearity is introduced, and indeed a cornerstone of modern deep learning (DL) (Hornik et al., 1989).

Despite their critical role, many of the most widely used AFs, such as the Rectified Linear Unit (ReLU) (Nair & Hinton, 2010), Leaky ReLU (LReLU) (Maas et al., 2013), SiLU (Elfwing et al., 2018) , and the Gaussian Error Linear Unit (GELU) (Hendrycks & Gimpel, 2016), are relatively simple, fixed, non-adaptive functions. While computationally efficient, their limited and static form of nonlinearity restricts the expressiveness of a network. To compensate, one approach is to build deeper or wider architectures, leading to increased computational costs, larger memory footprints, and more complex training procedures. Another approach is to adopt AFs which are trainable and/or have high degrees of nonlinearity. Examples of the latter include: PReLU (He et al., 2015), Swish (Ramachandran et al., 2017), CRReLU (Sun et al., 2025), PolyCom (Zhuo et al., 2025), and DiTAC (Chelly et al., 2024), leveraging more complex parameterizations and/or compositional forms. Although these methods introduce trainable parameters or compositional forms, their expressiveness remains constrained by predefined functional families. Furthermore, activation outputs from these AFs are typically information-theoretically incompressible, resulting in high inter-device communication costs in model-parallel settings. Thus, there is a pressing need for more powerful and flexible activation mechanisms that can endow neural networks with greater nonlinear capacity without relying solely on network depth or width.

In this paper, we propose a new family of trainable, architecture-agnostic AFs called Soft Quantization Activation Functions (SQUAFs). Unlike conventional AFs, SQUAFs are very flexible, trainable, and analytic, with closed-form formulas for computing partial derivatives with respect to both inputs and parameters. In particular, we show theoretically that SQUAFs can approximate any continuous nonlinear one-dimensional (1D) function with arbitrary precision, overcoming the inherent limitations of conventional AFs.

To validate the effectiveness of SQUAFs, we conduct extensive experiments across various model architectures, learning tasks and datasets. In function and image fitting tasks, multilayer perceptrons (MLPs) equipped with SQUAFs achieve dramatic error reduction and improvement of reconstruction quality, respectively, compared to their counterparts with conventional AFs, with gains up to $5000\times$ for the former and 25.27 dB for the latter. In image classification, SQUAFs work well for both convolutional neural networks (CNNs) and transformer-based networks, yielding consistent and significant accuracy gains over conventional AFs compared on CIFAR-100 and ImageNet-1K. In large language model (LLM) fine-tuning, we fine-tune GPT-2 (Radford et al., 2019) with SQUAFs using direct preference optimization (DPO) (Rafailov et al., 2023), getting consistent improvements in human preference alignment over the baseline. In knowledge distillation (KD), it is demonstrated that more than 3 times smaller distilled students with SQUAFs outperform their respective teachers with conventional AFs. Furthermore, we demonstrate that SQUAFs can reduce the inter-device communication cost in model-parallel settings by up to $9.4\times$ while still improving accuracy, paving the way for more scalable and efficient distributed training and inference. Our findings position SQUAFs as a simple yet powerful drop-in replacement for standard AFs, offering a path toward more expressive and efficient DL models.

Our main contributions are summarized as follows:

- We propose SQUAFs, a novel family of trainable, architecture-agnostic AFs for DL.

- We provide theoretical results showing that SQUAFs can approximate any continuous 1D function with arbitrary precision, with closed-form derivatives for efficient training.

- We empirically validate SQUAFs across diverse model architectures (MLPs, CNNs, vision and language transformers) and tasks (function and image fitting, image classification, LLM finetuning, etc), consistently demonstrating large performance gains.

- We show additional benefits of SQUAFs in model compression and inter-device communication cost reduction in model-parallel settings.

## 2 BACKGROUND AND RELATED WORK

### 2.1 ACTIVATION FUNCTIONS

Early AFs adopted in DNNs are sigmoidal, e.g., logistic and Tanh, functions. These functions, however, have vanishing gradients. To alleviate vanishing gradients, $\text{ReLU}(x) = \max(0, x)$ was proposed. ReLU is cheap to compute, and more crucially, does not face the vanishing-gradient issue. Due to these advantages, ReLU has been used predominately in DNNs for more than a decade. However, ReLU is a simple piecewise-linear function and offers limited nonlinearity, often necessitating large, complex DNN architectures with many layers of neurons in order to capture complex nonlinear relationships.

To increase the degree of nonlinearity in AFs, Leaky ReLU (LReLU), SiLU, and GELU were then proposed. Specifically, LReLU improves upon ReLU by allowing a small, non-zero slope (a "leak") for negative inputs, rather than outputting zero, thus maintaining a small gradient and preventing neurons from becoming inactive during training. SiLU, defined as $\text{SiLU}(x) = x\sigma(x)$, where $\sigma(x)$ is the logistic function, and GELU, defined as $\text{GELU}(x) = x\Phi(x)$, where $\Phi(x)$ is the cumulative distribution function of the standard normal distribution, are smooth variants of ReLU, effectively removing the nondifferentiability at the origin and further increasing nonlinearity around the origin. Since GPT-1 (Radford et al., 2018), GELU has been dominant in modern transformer-based models, e.g., GPT-2 (Radford et al., 2019), BERT (Devlin et al., 2019) and ViT (Dosovitskiy et al., 2020). On the other hand, SiLU has become dominant in UNet-based modern diffusion models (Ho et al., 2020; Rombach et al., 2022).

AFs mentioned above are all fixed and non-adaptive. To provide task-adaptive nonlinearity, trainable AFs have also been explored in the literature. For instance, PReLU is a trainable variant of LReLU where the slope of the negative part is optimized during training. Likewise, SiLU has been generalized to Swish, where a trainable temperature is inserted into the logistic function. More recently, CRReLU adds a regularization term to ReLU with a trainable weighting factor guided by an entropy-based AF optimization methodology. In general, trainable AFs like PReLU, Swish and CRReLU only moderately extend their non-trainable baselines due to simple parametrization. Therefore, to achieve even higher degrees of nonlinearity, recent studies on trainable AFs have focused on more complex functions with increased numbers of parameters and diverse functional classes. Zhuo et al. (2025) proposed polynomial composition activation functions (PolyCom), including PolyReLU and PolyNorm, which embed ReLU or normalization operations into weighted sum of polynomial functions. Chelly et al. (2024) introduced DiTAC, a diffeomorphism-based trainable AF using continuous piecewise-affine-based transformations. These trainable AFs take much more flexible shapes compared to earlier AFs. Nevertheless, they remain tied to a predefined functional family, limiting their expressiveness to some fixed patterns.

In contrast, kernel-based activation functions (KAFs) considered in (Scardapane et al., 2019), can approximate any continuous function over a subset of the real line. However, in practice, KAFs face many issues such as bias shift, vanishing gradient, and neuron death during training (Kiliçarslan & Celik, 2022). In addition, activation values from KAFs and the above AFs are generally incompressible in an information-theoretic sense, yielding high inter-device communication costs in model-parallel settings. These challenges underscore the need for activation mechanisms that are simultaneously universal, trainable, analytically tractable, and communication-efficient.

## 2.2 SCALAR QUANTIZERS

A scalar quantizer is a function $Q$ that maps each $x \in \mathbb{R}$ into a finite reproduction alphabet

$$\hat{\mathcal{A}} = \{y_1, y_2, \cdots, y_L\} \subset \mathbb{R}.$$

When $Q(x) = y_i$, $x$ is said to be quantized to $y_i$. The quantizer $Q$ is said to be uniform if

$$\hat{\mathcal{A}} = \{0, \pm q, \pm 2q, \ldots, \pm kq\}$$

and each $x$ is quantized to the nearest point in $\hat{\mathcal{A}}$, where $k$ is a positive integer and $q$ is a positive real number called the quantization step size.

## 2.3 PROBABILISTIC AND SOFT DETERMINISTIC QUANTIZERS

Let $P(\cdot|x)$ be a conditional probability mass function (CPMF) over $\hat{\mathcal{A}}$ or equivalently its index set $\mathcal{A} = \{1, 2, \cdots, L\}$ (or $\{0, \pm 1, \pm 2, \ldots, \pm k\}$ in the case of uniform quantization), given $x \in \mathbb{R}$. That is, for each $x \in \mathbb{R}$, $P(y_i|x) \geq 0, \forall y_i \in \hat{\mathcal{A}}$ and $\sum_{i=1}^{L} P(y_i|x) = 1$. With $P(\cdot|x)$, each $x$ can be quantized randomly to $y_i \in \hat{\mathcal{A}}$ with probability $P(y_i|x)$. Denote this random mapping by $Q_p(x)$, which is a probabilistic quantizer.

Given $x$, $Q_p(x)$ is a random variable taking values in $\hat{\mathcal{A}}$. Consider the expectation of $Q_p(x)$:

$$Q_d(x) = \mathbb{E}[Q_p(x)] = \sum_{i=1}^{L} y_i \cdot P(y_i|x). \tag{1}$$

$Q_d(x)$ is the soft deterministic quantizer corresponding to $Q_p(x)$. $Q_d(x)$ and $Q_p(x)$ were introduced by Yang & Hamidi (2025) to formulate coded deep learning.

Many types of CPMFs can be used for $Q_d(x)$ and $Q_p(x)$. Let $\alpha > 0$ be a trainable parameter. The following are two well-known examples.

**Example 1**: For any $x$,

$$P(y_i|x) = \frac{e^{-\alpha(x-y_i)^2}}{\sum_{j=1}^{L} e^{-\alpha(x-y_j)^2}}, \tag{2}$$

which is the discrete counterpart of a conditional Gaussian distribution.

**Example 2**: For any $x$,

$$P(y_i|x) = \frac{e^{-\alpha|x-y_i|}}{\sum_{j=1}^{L} e^{-\alpha|x-y_j|}}, \tag{3}$$

which is the discrete counterpart of a conditional Laplacian distribution.

## 2.4 STEP FUNCTIONS

Let $\{C_i\}_{i=1}^{L}$ be an interval partition of the whole real line $\mathbb{R}$. That is, each $C_i$ is an interval; they are pairwise disjoint, $C_i \cap C_j = \emptyset, \forall i \neq j$; and together they cover the whole real line, $\cup_{i=1}^{L} C_i = \mathbb{R}$. A step function $f(x)$ can be written as

$$f(x) = \sum_{i=1}^{L} z_i \mathcal{X}_{C_i}(x), \tag{4}$$

where $\mathcal{X}_C(x)$ is the indicator function of the set $C$, which is equal to 1 if $x \in C$ and 0 otherwise.

Compared with scalar quantizers, the step function $f(x)$ in equation 4 can also be regarded as a generalized scalar quantizer which maps $x$ to $z_i$ if and only if $x \in C_i$. The difference between the step function $f(x)$ and the scalar quantizer $Q$ lies in that $x$ and $Q(x) = y_i$ generally belong to the same partition interval $\{y : Q(y) = y_i\}$, whereas $x$ and $f(x) = z_i$ have no such constraint.

# 3 SQUAFs

Inspired by probabilistic and soft deterministic quantizers, we now introduce general CPMFs into step functions, yielding our proposed soft quantization activation functions.

## 3.1 DEFINITIONS

Recall that $\hat{\mathcal{A}} = \{y_1, y_2, \cdots, y_L\} \subset \mathbb{R}$. For convenience, we also regard $\hat{\mathcal{A}}$ as a vector of dimension $L$ denoted by $\boldsymbol{y} := [y_i]_{i=1}^{L}$. Likewise, write $\boldsymbol{z} := [z_i]_{i=1}^{L}$. Based on $\boldsymbol{y}$ and a general CPMF $P(\cdot|x)$ over $\hat{\mathcal{A}}$, we replace the indicator functions $\mathcal{X}_{C_i}(x)$ in equation 4 with $P(y_i|x)$, resulting in a soft quantization activation function (SQUAF)

$$\phi(x) = \sum_{i=1}^{L} z_i \cdot P(y_i|x). \tag{5}$$

Corresponding to $\phi(x)$, there is a probabilistic quantization activation function (SQUAF-P) $\phi_p(x)$ that maps each $x$ randomly to $z_i$ with probability $P(y_i|x)$. The relationship between $\phi_p(x)$ and $\phi(x)$ is analogous to that between $Q_p(x)$ and $Q_d(x)$. $\phi_p(x)$ is a probabilistic step function; being the expectation of $\phi_p(x)$, $\phi(x)$ is its corresponding soft deterministic step function.

**Example 3**: When the CPMF $P(\cdot|x)$ is the discrete conditional Gaussian distribution in Example 1, $\phi(x)$ can be conveniently expressed as

$$\phi(x) = \phi(x|\boldsymbol{y}, \boldsymbol{z}, \alpha) = \sum_{i=1}^{L} z_i \frac{e^{-\alpha(x-y_i)^2}}{\sum_{j=1}^{L} e^{-\alpha(x-y_j)^2}} \tag{6}$$

in the general case, and as

$$\phi(x) = \phi(x|q, \boldsymbol{z}, \alpha) = \sum_{i=-k}^{k} z_i \frac{e^{-\alpha(x-iq)^2}}{\sum_{j=-k}^{k} e^{-\alpha(x-jq)^2}} \tag{7}$$

when $\hat{\mathcal{A}} = \{0, \pm q, \pm 2q, \ldots, \pm kq\}$.

**Remark.** (1) $\phi(x)$ in Example 3 is differentiable with respect to $(x, \boldsymbol{y}, \boldsymbol{z}, \alpha)$ or $(x, q, \boldsymbol{z}, \alpha)$ as the case maybe. The analytical partial derivatives are presented in Appendix A.1. (2) With proper choices of $(\boldsymbol{y}, \boldsymbol{z}, \alpha)$ or $(q, \boldsymbol{z}, \alpha)$ in Example 3, $\phi(x)$ does not have vanishing gradients. (3) The role of $\boldsymbol{z}$ is to approximate the amplitude of a desirable nonlinear activation function, while the role of $\boldsymbol{y}$ or $q$ in Example 3 is to partition the domain of the desirable nonlinear activation function. (4) $(\boldsymbol{y}, \boldsymbol{z}, \alpha)$ or $(q, \boldsymbol{z}, \alpha)$ in Example 3 are trainable parameters, and $\alpha$ can be regarded as a horizontal scaling factor.

### 3.2 Universal Approximation

As illustrated in Example 3, let us introduce a horizontal scaling factor $\alpha$ into the general CPMF $P(\cdot|x)$. Write $P(\cdot|x)$ as $P_\alpha(\cdot|x)$ and $\phi(x)$ in equation 5 as $\phi(x|\boldsymbol{y}, \boldsymbol{z}, \alpha)$ to explicitly show dependence on $\alpha$. Assume $L \geq 2$. For each $x$, let $y_{i(x)}$ and $y_{j(x)}$ be the two nearest points of $x$ in $\hat{\mathcal{A}}$. In order for $\phi(x|\boldsymbol{y}, \boldsymbol{z}, \alpha)$ to facilitate the learning process during DNN training and be a universal approximator for 1D functions, we further assume that the CPMF $P_\alpha(\cdot|x)$ over $\hat{\mathcal{A}}$ satisfies the following properties:

**P1** $P_\alpha(\boldsymbol{y}|x) = [P_\alpha(y_1|x), P_\alpha(y_2|x), \cdots, P_\alpha(y_L|x)]$, as a function of $\boldsymbol{y}$ and $x$, is differentiable almost everywhere.

**P2** As $\alpha$ goes to $\infty$, $\sup_x \min_{0 \leq \lambda \leq 1} |\phi(x|\boldsymbol{y}, \boldsymbol{z}, \alpha) - \lambda z_{i(x)} - (1 - \lambda)z_{j(x)}| \to 0$.

Property P2 is a localization property. It implies that when $\alpha$ is large, $\phi(x)$ depends mainly on $z_{i(x)}$ and $z_{j(x)}$. In other words, each amplitude value in $\boldsymbol{z}$ has mainly a local effect on the shape of $\phi(x)$, which is advantageous during optimization of $\boldsymbol{z}$ in the training process. With Properties P1 and P2, $\phi(x)$ can approximate any continuous nonlinear function on a closed interval with arbitrary precision. We formalize this in the following theorem, which is proved in Appendix A.2.

**Theorem 1** (Universal approximation). *Assume that the CPMF $P_\alpha(\cdot|x)$ has Properties P1 and P2. Let $g$ be any continuous function defined on a close interval $[a, b]$. For every $\varepsilon > 0$, there exist $L \in \mathbb{N}$, and $(\boldsymbol{y}, \boldsymbol{z}, \alpha)$ such that the SQUAF $\phi(x)$ as in equation 5 satisfies $\sup_{x \in [a,b]} |\phi(x) - g(x)| \leq \varepsilon$.*

Many types of CPMFs satisfy Properties P1 and P2. In particular, the discrete conditional Guassian and Laplacian distributions in Examples 1 and 2 have Properties P1 and P2, as shown in the following theorem, which is proved in Appendix A.2 as well.

**Theorem 2.** *If $P_\alpha(\cdot|x)$ is given by equation 2 or equation 3, then it satisfies Properties P1 and P2.*

### 3.3 Integration of SQUAFs to DNNs

We abstract a DNN with any architecture into a model with $T$ hidden transforms, separated by $T-1$ AFs. Formally, a DNN $f_\Theta : \mathbb{R}^n \to \mathbb{R}^m$ can be expressed as

$$f_\Theta(\boldsymbol{u}) = (h_T \circ \varphi_{T-1} \circ h_{T-1} \circ \cdots \circ \varphi_1 \circ h_1)(\boldsymbol{u}), \tag{8}$$

where $\Theta$ denotes the trainable parameters of the DNN, $\boldsymbol{u}$ denotes the DNN's input, $\{\varphi_1, \ldots, \varphi_{T-1}\}$ are AFs, and $\{h_1, \ldots, h_T\}$ are the hidden transforms. Each hidden transform potentially includes modules such as fully connected layers, convolutional layers, attention layers, normalization layers, etc. Denote activations as $\boldsymbol{a}_t = \varphi_t(\boldsymbol{x}_t)$ for $t = 1, 2, \ldots, T-1$, where $\varphi_t$ is applied to pre-activations $\boldsymbol{x}_t$ in an element-wise manner.

We incorporate SQUAFs into a DNN by setting

$$\varphi_t(\boldsymbol{x}) = \phi(\boldsymbol{x}|\boldsymbol{y}_t, \boldsymbol{z}_t, \alpha_t), \ \ t = 1, 2, \ldots, T-1, \tag{9}$$

where $\boldsymbol{y}_t := [y_i^t]_{i=1}^L$, $\boldsymbol{z}_t := [z_i^t]_{i=1}^L$, and $\alpha_t$ are the parameters in the $t$-th SQUAF, while we incorporate SQUAF-Ps into a DNN by setting

$$\varphi_t(\boldsymbol{x}) = \text{sg}\left[\phi_p(\boldsymbol{x}|\boldsymbol{y}_t, \boldsymbol{z}_t, \alpha_t) - \phi(\boldsymbol{x}|\boldsymbol{y}_t, \boldsymbol{z}_t, \alpha_t)\right] + \phi(\boldsymbol{x}|\boldsymbol{y}_t, \boldsymbol{z}_t, \alpha_t), \ \ t = 1, 2, \ldots, T-1, \tag{10}$$

where $\text{sg}[\cdot]$ stands for the stop-gradient operator defined as identity in the forward pass and has zero partial derivatives. In the latter case, SQUAF-P is used in the forward pass to produce discrete activations, while SQUAF serves as its gradient proxy in backpropagation.

To train a DNN equipped with SQUAF or SQUAF-P, we solve the following minimization problem

$$\min_\Theta \mathbb{E}\left[\mathcal{J}\left(f_\Theta, \boldsymbol{v}\right) + \lambda \mathcal{R}(\boldsymbol{a})\right], \tag{11}$$

where $\boldsymbol{v}$ denotes the data, $\mathcal{J}$ is a generalized loss function used in supervised, unsupervised or other types of learning such as cross-entropy loss, mean squared error (MSE) and language modeling loss, $\boldsymbol{a} := \{\boldsymbol{a}_t\}_{t=1}^{T-1}$ denotes the collection of all activations, $\mathcal{R}$ denotes the rate of activations in bit-per-activation (bpa), and $\lambda$ is a hyperparameter. In general, $\Theta = \{W, (\boldsymbol{y}_1, \boldsymbol{z}_1, \alpha_1), \ldots, (\boldsymbol{y}_{T-1}, \boldsymbol{z}_{T-1}, \alpha_{T-1})\}$, where $W$ denotes all trainable parameters in $\{h_t\}_{t=1}^T$.

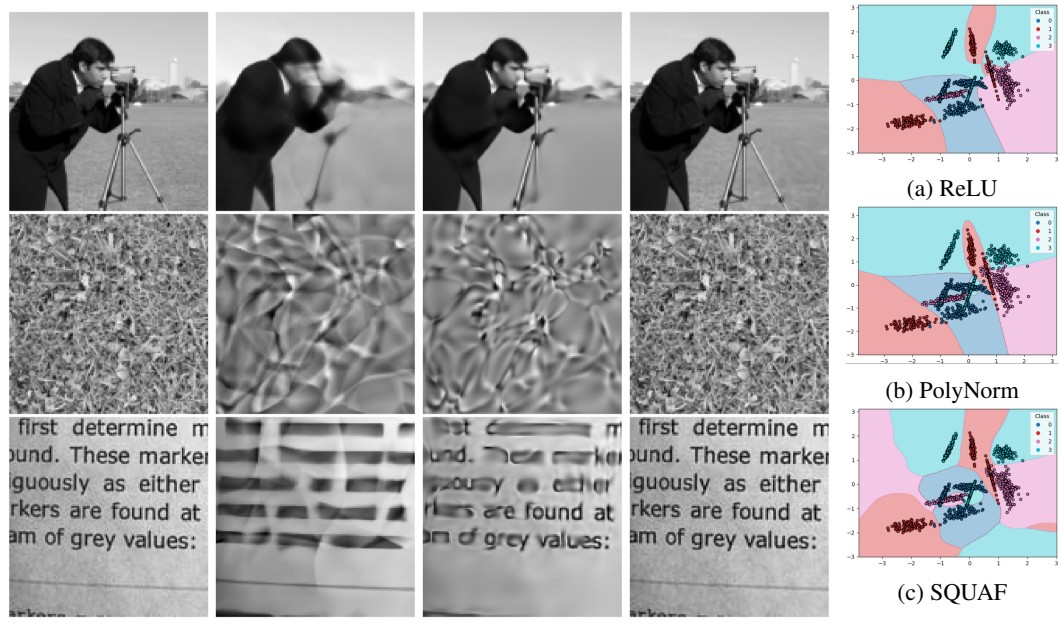

(a) ReLU

(b) PolyNorm

(c) SQUAF

Ground Truth   ReLU   PolyNorm   SQUAF

Figure 1: Ground truth images of Camera, Grass and Page (from top to bottom) are displayed in the leftmost column, followed by images fitted by MLPs equipped with ReLUs, PolyNorms and SQUAFs.

Figure 2: Decision boundary visualization for the toy classification task with $C = 4$.

However, in some cases, if the initialization of some $y$, $z$ or $\alpha$ is good enough, they can be excluded from $\Theta$. Note that for a SQUAF-DNN, i.e., a DNN equipped with SQUAF following eq. 8 and 9, $\lambda$ is always set to 0 since each activation value is essentially analog and requires 32 bits to represent in the case of full precision and hence $\mathcal{R}(\boldsymbol{a})$ is a constant; for a SQUAF-P-DNN, i.e., a DNN equipped with SQUAF-P following eq. 8 and 10, $\lambda$ can be positive in order to improve activation compression. In the latter case, $\mathcal{R}(\boldsymbol{a})$ is a function of $\Theta$, expressed by

$$\mathcal{R}(\boldsymbol{a}) = \frac{\sum_{t=1}^{T-1} |\boldsymbol{a}_t| \cdot r(\boldsymbol{a}_t)}{\sum_{t=1}^{T-1} |\boldsymbol{a}_t|}, \; r(\boldsymbol{a}_t) = H\left(P(a_t)\right), \; P(a_t) = \frac{1}{|\boldsymbol{x}_t|} \sum_{l=1}^{|\boldsymbol{x}_t|} P(a_t|\boldsymbol{x}_t[l]), \; \forall a_t \in \{z_i^t\}_{i=1}^{L}, \quad (12)$$

where $|\cdot|$ denotes the cardinality of a vector, $|\boldsymbol{a}_t| = |\boldsymbol{x}_t|$, $\boldsymbol{x}_t[l]$ denotes the $l$-th element of $\boldsymbol{x}_t$, $P(z_i^t|\boldsymbol{x}_t[l]) = P(y_i^t|\boldsymbol{x}_t[l])$, $P(a_t)$ is the marginal probability mass function (MPMF) of activations in $\boldsymbol{a}_t$, and $H(\cdot)$ denotes the Shannon entropy of a probability distribution. Note that Shannon entropy is an analytical proxy of the actual rate resulting from entropy coding, e.g., Huffman coding, and thus $R(\boldsymbol{a})$ is a close approximation of the average number of bits to represent an activation. The above loss function is inspired by coded deep learning (Yang & Hamidi, 2025).

## 4 EXPERIMENTS

Our results in this section are obtained by $\phi(x)$ in the form of equation 7 or its corresponding $\phi_p(x)$.

### 4.1 REGRESSION

**Function fitting.** We consider fitting two sinusoidal functions of increasing complexity. The first is a 1D function: $g(x) = 0.4\sin(19x) + 0.2\sin(23x) + 0.3\sin(29x) + 0.1\sin(31x)$ within domain $[-1, 1]$, and the second is a 2D function: $g(x_1, x_2) = 0.4\sin(9x_1 - 7x_2) + 0.1\sin(-9x_1 + 11x_2) + 0.15\sin(3x_1 + 13x_2) + 0.15\sin(9x_1 + 9x_2) + 0.1\sin(13x_1 + 5x_2) + 0.1\sin(3x_1 + 19x_2)$ within domain $[-1, 1]^2$. We use a $[1, 64, 1]$ MLP (i.e., a three-layer MLP with one input neuron, 64 hidden neurons, and one output neuron) for the 1D case, and a $[2, 50, 1]$ MLP for the 2D case, both equipped with SQUAF or other compared AFs. For SQUAF, we choose $k = 2$, so only a negligible number of parameters are added to the MLP. The MLP is trained for 40000 iterations using an Adam optimizer (Kingma & Ba, 2014) following the recipe in Chelly et al. (2024). Table 3 shows

Table 1: Results for image fitting.

| AF | Camera | | Grass | | Page | |
|---|---|---|---|---|---|---|
| | PSNR (dB) | SSIM | PSNR (dB) | SSIM | PSNR (dB) | SSIM |
| ReLU | 26.75 | 0.8172 | 22.70 | 0.4742 | 19.65 | 0.6223 |
| GELU | 28.23 | 0.8496 | 22.06 | 0.3813 | 19.42 | 0.5868 |
| LReLU | 27.27 | 0.8270 | 22.86 | 0.4903 | 20.05 | 0.6510 |
| PReLU | 26.54 | 0.8158 | 22.87 | 0.4932 | 20.05 | 0.6603 |
| Swish | 27.36 | 0.8238 | 21.55 | 0.3140 | 19.21 | 0.5469 |
| CRReLU | 27.47 | 0.8403 | 22.83 | 0.4855 | 20.07 | 0.6650 |
| PolyReLU | 25.98 | 0.7986 | 23.20 | 0.5438 | 19.72 | 0.6258 |
| PolyNorm | 31.76 | 0.8900 | 23.34 | 0.5426 | 21.58 | 0.7195 |
| DiTAC | 27.84 | 0.8409 | 21.10 | 0.2529 | 19.39 | 0.5857 |
| SQUAF | **43.20** | **0.9786** | **48.61** | **0.9991** | **45.25** | **0.9941** |

Table 2: Results for toy classification.

| AF | $C = 3$ | | $C = 4$ | |
|---|---|---|---|---|
| | Param. | Acc. (%) | Param. | Acc. (%) |
| ReLU | 4547 | 93.89 | 4612 | 70.42 |
| GELU | 4547 | 91.11 | 4612 | 72.50 |
| LReLU | 4547 | 93.33 | 4612 | 73.75 |
| PReLU | 4549 | 93.89 | 4614 | 73.75 |
| Swish | 4549 | 92.22 | 4614 | 60.42 |
| CRReLU | 4549 | 93.89 | 4614 | 74.58 |
| PolyReLU | 4555 | 96.11 | 4620 | 78.75 |
| PolyNorm | 4555 | 96.67 | 4620 | 85.00 |
| DiTAC | 4565 | 92.78 | 4630 | 74.17 |
| SQUAF | 4561 | **97.22** | 4626 | **93.75** |

the result of SQUAF compared to other AFs. Clearly, SQUAF achieves the supreme performance among compared AFs with comparable number of trainable parameters, reducing MSE by 1~4 orders of magnitude and reaching almost perfect $R^2$ score. Note that KAF is originally an element-dependent AF, i.e., each hidden neuron is connected to a KAF with distinct parameterization. In our experiments, we make it element-independent as a conventional AF so that its number of trainable parameters is comparable to other AFs. In this case, its performance is generally the worst among all compared AFs. Therefore, we exclude KAF from subsequent experimental comparisons. Additionally, for the 1D case, we conduct experiments where only a SQUAF is used to fit the function, getting rid of the two fully connected layers in the MLP. In this case, we try to either match the number of trainable parameters in SQUAF to that of the baseline MLP or match the performance to that of the best compared AF (DiTAC). In the former case, SQUAF gets an extra 200× error reduction compared to the MLP equipped with SQUAF, and in the latter case, the number of trainable parameters is reduced dramatically, achieving a 7.2× compression ratio compared to DiTAC. The success of this set of experiments is attributed to the universal approximation capability of SQUAF, and demonstrates its potential in Kolmogorov–Arnold networks (KANs) (Liu et al., 2025) for tasks such as partial differential equation (PDE) solving. We will further investigate this in future work.

**Image fitting.** This task treats pixel coordinates as inputs to an MLP, which is trained to output the corresponding pixel intensities (Stanley, 2007; Tancik et al., 2020). In other words, the goal is to represent/memorize an image by learning a coordinate-to-intensity mapping. For grayscale images, this can be regarded as a special case of 2D function fitting, where the target function is not analytic but implicitly defined by the image itself. We consider fitting three images from van der Walt et al. (2014), namely Camera, Grass and Page, shown in Fig. 1. We train a $[2, 128, 128, 128, 128, 1]$ MLP with Adam optimizer for 1000 epochs to fit each target image. Numerical results are presented in Tab. 1, where SQUAF is shown to dramatically improve the image reconstruction quality, up to a 25.27 dB PSNR gain or a 0.4553 SSIM gain over a runner-up AF. We also compare fitted images to the ground truth images visually in Fig. 1. It's shown that SQUAF can fit all three ground truth images almost flawlessly, while the baseline ReLU and runner-up PolyNorm give highly blurry results, especially in Grass and Page, where high-frequency components prevail. The success of SQUAF in image fitting can be potentially extended to neural radiance field (NeRF) (Mildenhall et al., 2021), in which MLPs are used to fit 3D scenes, and/or other tasks involving the modeling of high-frequency signals. We will investigate this direction in future work.

## 4.2 CLASSIFICATION

**Toy classification.** Before conducting experiments on standard classification datasets, we demonstrate the classification capability of SQUAF-DNN with a proof-of-concept experiment on a toy dataset. To this end, we construct a 2D Gaussian-Mixture-Model (GMM) with $C$ classes, each class having 3 clusters with 100 samples for each cluster. The total $300C$ samples are split into training and test sets with ratio $4 : 1$. For this task, we train a $[2, 64, 64, C]$ MLP with Adam optimizer for 100 epochs. Top-1 test accuracy and number of trainable parameters for each AF is shown in Tab. 2 for $C = 3$ and $C = 4$. In $C = 3$ case, all AFs achieve decent results while SQUAF being the best with a marginal gain. However, in $C = 4$ case, all compared AFs become significantly worse, while SQUAF can still get decent accuracy, outperforming the runner-up (PolyNorm) by 8.75%. This happens because SQUAF can provide more nonlinear decision boundaries. In $C = 3$ case, clusters are relatively easy to separate, so simple decision boundaries with low degrees of nonlinearity can succeed with high probability. However, in $C = 4$ case, clusters are highly interleaving to the extend that only highly nonlinear decision boundaries can separate them. See Fig. 2 (best viewed in

Table 3: Results for function fitting. We report the number of trainable parameters, MSE, and $R^2$ score for each model. In the last two rows, we report results for 1D function fitting solely based on SQUAFs. We highlight the best results in bold, and the second best results with underscores. More implementation details can be found in Appendix A.3.

| AF | Param. | 1D Function MSE↓ | $R^2$ ↑ | Param. | 2D Function MSE↓ | $R^2$ ↑ |
|---|---|---|---|---|---|---|
| ReLU | 193 | $1.0 \times 10^{-2}$ | 93.3 | 201 | $2.3 \times 10^{-2}$ | 88.9 |
| GELU | 193 | $1.1 \times 10^{-1}$ | 9.7 | 201 | $5.4 \times 10^{-2}$ | 65.3 |
| LReLU | 193 | $8.0 \times 10^{-3}$ | 94.7 | 201 | $1.1 \times 10^{-2}$ | 93.1 |
| PReLU | 194 | $1.0 \times 10^{-2}$ | 93.1 | 202 | $8.0 \times 10^{-3}$ | 96.1 |
| Swish | 194 | $4.0 \times 10^{-3}$ | 97.5 | 202 | $1.3 \times 10^{-1}$ | 25.5 |
| CRReLU | 194 | $9.7 \times 10^{-3}$ | 93.7 | 202 | $1.3 \times 10^{-2}$ | 90.4 |
| PolyReLU | 197 | $7.1 \times 10^{-3}$ | 95.1 | 205 | $8.1 \times 10^{-3}$ | 93.4 |
| PolyNorm | 197 | $6.4 \times 10^{-2}$ | 56.9 | 205 | $7.0 \times 10^{-2}$ | 39.9 |
| DiTAC | 202 | $1.0 \times 10^{-3}$ | 99.3 | 210 | $4.0 \times 10^{-3}$ | 98.4 |
| KAF | 200 | $1.2 \times 10^{-1}$ | 1.9 | 208 | $1.1 \times 10^{-1}$ | 6.4 |
| SQUAF | 200 | $4.0 \times 10^{-5}$ | 100.0 | 208 | $4.0 \times 10^{-4}$ | 99.7 |
| SQUAF-only | 192 | $2.0 \times 10^{-7}$ | 100.0 | n/a | n/a | n/a |
| SQUAF-only | 28 | $1.0 \times 10^{-3}$ | 99.3 | n/a | n/a | n/a |

Table 4: Top-1 validation accuracy (%) on ImageNet-1K.

| AF | ResNet-18 69.76 | ResNet-34 73.31 | ViT-T 73.90 | Swin-T 81.28 |
|---|---|---|---|---|
| ReLU | 69.76 (+0.00) | 73.31 (+0.00) | 73.43 (-0.47) | 80.70 (-0.58) |
| GELU | 70.66 (+0.90) | 73.71 (+0.40) | 73.90 (+0.00) | 81.28 (+0.00) |
| SiLU | 70.51 (+0.75) | 72.76 (-0.55) | 73.70 (-0.20) | 80.29 (-0.99) |
| SQUAF | **70.88** (+1.12) | **74.44** (+1.13) | **75.00** (+1.10) | **81.77** (+0.49) |

Table 5: Comparison of performance metrics between SQUAF and GELU for GPT-2 fine-tuning based on DPO.

| | | ERA (%) ↑ | ERM ↑ | EL ↓ |
|---|---|---|---|---|
| $\beta = 0.1$ | GELU | 62.11 | 0.0977 | 0.6646 |
| | SQUAF | **62.50** (+0.39) | **0.1352** | **0.6510** |
| $\beta = 0.2$ | GELU | 59.76 | 0.1639 | 0.6733 |
| | SQUAF | **62.89** (+3.13) | **0.1838** | **0.6544** |
| $\beta = 0.4$ | GELU | 58.59 | 0.3033 | 0.7184 |
| | SQUAF | **61.33** (+2.74) | **0.4201** | **0.6547** |

color and zoomed in) for a visual demonstration of decision boundaries learned by baseline ReLU, runner-up PolyNorm and SQUAF in $C = 4$ case. For the pink cluster in the middle of the plot, only SQUAF can correctly classify most samples in it by carving a "hole" inside the blue decision region, demonstrating its capability of learning complex decision boundaries.

**CIFAR-100.** We perform experiments across 6 model architectures, namely ShuffleNetV1 (Zhang et al., 2018), VGG-8, VGG-13 (Simonyan & Zisserman, 2014), ResNet-56 (He et al., 2016), ViT-T (Touvron et al., 2021a) and EfficientFormer-L1 (Li et al., 2022), with the first 4 being CNNs and the last 2 being transformer-based networks. For all experiments on CNNs, we follow the training recipe proposed by Tian et al. (2019), and for those on transformer-based networks, we follow the training recipe proposed by Xu et al. (2023). See Appendix A.3 for implementation details. Results on CIFAR-100 are shown in Tab. 6, where SQUAF consistently outperforms all compared AFs, achieving up to 2.24% accuracy gain.

Table 6: Top-1 validation accuracy (%) on CIFAR-100. Results are averaged over 3 runs. The numbers in parentheses show the performance differences between each AF and the baseline model, with green denoting outperformance and red denoting underperformance. If a model diverges or runs into numerical issues during training, we denote its performance with "–". Note that the original EfficientFormer-L1 uses hybrid AFs, so its baseline accuracy doesn't align with any AF.

| AF | ShuffleNetV1 70.50 | VGG-8 70.36 | ResNet-56 72.34 | VGG-13 74.64 | ViT-T 71.84 | EfficientFormer-L1 80.26 |
|---|---|---|---|---|---|---|
| ReLU | 70.50 (+0.00) | 70.36 (+0.00) | 72.34 (+0.00) | 74.64 (+0.00) | 72.06 (+0.22) | 80.67 (+0.41) |
| GELU | 71.28 (+0.78) | 69.94 (-0.42) | 72.45 (+0.11) | 74.53 (-0.11) | 71.84 (+0.00) | 80.86 (+0.60) |
| LReLU | 71.51 (+1.01) | 70.48 (+0.12) | 72.71 (+0.37) | 75.00 (+0.36) | 71.43 (-0.41) | 80.52 (+0.26) |
| PReLU | 70.38 (-0.12) | 68.30 (-2.06) | 71.76 (-0.58) | 73.33 (-1.31) | 72.06 (+0.22) | 79.76 (-0.50) |
| SiLU | 72.08 (+1.58) | 70.28 (-0.08) | 72.16 (-0.18) | 74.68 (+0.04) | 71.68 (-0.16) | 80.33 (+0.07) |
| Swish | 72.25 (+1.75) | 70.50 (+0.14) | 73.25 (+0.91) | 74.86 (+0.22) | 71.44 (-0.40) | 80.55 (+0.29) |
| CRReLU | 68.18 (-2.32) | 68.22 (-2.14) | 71.43 (-0.91) | 72.78 (-1.86) | 71.17 (-0.67) | 79.19 (-1.07) |
| PolyReLU | – | – | – | – | 70.66 (-1.18) | – |
| PolyNorm | 65.58 (-4.92) | 66.65 (-3.71) | 65.52 (-6.82) | 68.53 (-6.11) | 71.44 (-0.40) | 79.53 (-0.73) |
| DiTAC | 71.29 (+0.79) | 71.03 (+0.67) | 73.22 (+0.88) | 74.52 (-0.12) | 4.35 (-67.49) | 34.41 (-45.85) |
| SQUAF | **72.74** (+2.24) | **72.57** (+2.21) | **73.93** (+1.59) | **76.05** (+1.41) | **73.15** (+1.31) | **81.48** (+1.22) |

**ImageNet-1K.** We perform experiments across 4 model architectures, i.e., ResNet-18, ResNet-34, ViT-T and Swin-T (Liu et al., 2021), with the first 2 being CNNs and the last 2 being transformer-based networks. For CNNs, we follow the standard PyTorch (Paszke et al., 2019) training recipe, and for transformer-based networks, we follow the training recipe proposed by Xu et al. (2023). Results on ImageNet-1K are shown in Tab. 4, where SQUAF consistently improves over the baseline accuracy, achieving up to 1.13% gain. Please refer to Appendix A.10 for visualization of SQUAFs.

## 4.3 LLM Fine-tuning

Using DPO, we fine-tune GPT-2 (124M parameters) on Stanford Human Preferences dataset (Ethayarajh et al., 2022) and Anthropic HH dataset (Bai et al., 2022). Following the recipe given by

Rafailov et al. (2023), the pretrained GPT-2 first goes through a supervised fine-tuning (SFT) stage to obtain a reference policy, and then a DPO stage for aligning with human preference. SQUAFs are embedded into GPT-2 in both stages. As suggested in Rafailov et al. (2023), we select the hyperparameter $\beta$ in DPO loss to be within $[0.1, 0.5]$, and compare SQUAF to the baseline GELU in GPT-2. We report evaluation reward accuracy (ERA), evaluation reward margin (ERM) and evaluation loss (EL) as performance metrics in Tab. 5, where SQUAF is shown to consistently outperform GELU.

Table 7: Top-1 validation accuracy (%) on CIFAR-100 and the number of trainable parameters in a model. Results are averaged over 3 runs.

| Teacher | VGG-13 | | WRN-40-2 | |
|---|---|---|---|---|
| | 74.64 | 9462180 | 75.61 | 2255156 |
| Student | VGG-8 | | WRN-16-2 | |
| ReLU | 72.98 | 3965028 | 74.92 | 703284 |
| GELU | 73.51 (+0.53) | 3965028 | 75.30 (+0.38) | 703284 |
| LReLU | 73.37 (+0.39) | 3965028 | 74.90 (-0.02) | 703284 |
| PReLU | 72.27 (-0.71) | 3965033 | 74.93 (+0.01) | 703297 |
| Swish | 73.45 (+0.47) | 3965033 | 75.36 (+0.44) | 703297 |
| CRReLU | 72.88 (-0.10) | 3965033 | 74.98 (+0.06) | 703297 |
| PolyReLU | – | 3965048 | – | 703336 |
| PolyNorm | 70.44 (-2.54) | 3965048 | 72.86 (-2.06) | 703336 |
| DiTAC | 73.49 (+0.51) | 3965073 | 75.45 (+0.53) | 703401 |
| SQUAF | **74.95** (+1.97) | 3965063 | **75.93** (+1.01) | 703319 |

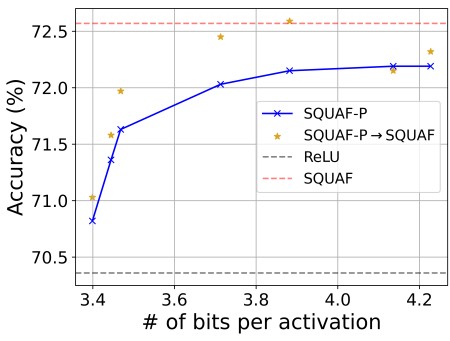

Figure 3: Rate-accuracy curve for SQUAF-P. SQUAF-P→SQUAF denotes switching SQUAF-P to SQUAF during inference.

### 4.4 EXTENSIONS

**Distilling knowledge into SQUAF-DNNs.** KD (Hinton et al., 2015; Zheng & YANG, 2024) is one of the most popular ways towards model compression. However, in most cases, a student model can only get close to its teacher's performance but cannot catch up with or surpass that. This is verified in Tab. 7, where student models with baseline AFs never outperform their respective teacher models. However, student models with SQUAFs are not only smaller than their respective teachers but also obtain higher classification accuracy, achieving up to $3.2\times$ compression ratio with nontrivial accuracy improvement. Experiments are carried out with VGGs and Wide ResNets (WRNs) (Zagoruyko & Komodakis, 2016) using the distillation recipe in Tian et al. (2019).

**Reducing communication cost with SQUAF-P.** We incorporate SQUAF-Ps into VGG-8 and train it on CIFAR-100 following eq. 11 with $\lambda \in \{0, 0.001, 0.002, \ldots, 0.006\}$. The resulting validation accuracy and rate of activations based on Huffman coding is illustrated in Fig. 3. It's shown that SQUAF-P can reduce the rate of activations from full precision (32 bpa) down to 3.4 bpa while improving the accuracy over the baseline ReLU. Interestingly, we find that if we replace pretrained SQUAF-Ps with SQUAFs after training without modifying the learned parameters $\{(q_t, z_t, \alpha_t)\}_{t=1}^{T-1}$, the model's accuracy is generally improved, up to the accuracy obtained by training VGG-8 with SQUAF. This property is beneficial in some scenarios. Large models can be trained in model-parallel setting, but deployed on a single device. In this case, one only cares about inter-device communication cost during training. Therefore, after the model is trained, one can simply switch SQUAF from the probabilistic mode to the deterministic mode during inference, resulting in improved accuracy, reduced latency and deterministic output. See Appendix A.4 for computational efficiency analysis.

## 5 CONCLUSION

The paper proposed SQUAF, a highly nonlinear, trainable, analytic, and compressible activation function that can approximate any continuous 1D function with arbitrary precision. Equipped with SQUAFs, DNNs can achieve much higher expressiveness than using other AFs and thus do better in all kinds of data modeling. The proposed SQUAF is experimented on numerous tasks, datasets and DNN models, where it consistently and significantly improves performance of DNNs. Overall, SQUAF is a simple yet effective AF agnostic to DNN architectural design.

## REPRODUCIBILITY STATEMENT

Please refer to Section 4 and Appendix A.3 to reproduce our experimental results.

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

## A  APPENDIX

### A.1  PARTIAL DERIVATIVES OF SQUAF

We now compute the partial derivatives of $\phi(x)$ w.r.t. each component of $(x, \boldsymbol{y}, \boldsymbol{z}, \alpha)$ or $(x, q, \boldsymbol{z}, \alpha)$ when $P(\cdot|x)$ is the discrete conditional Gaussian distribution in Example 1.

For $x$, we have

$$\frac{\partial}{\partial x} P(y_i|x) = P(y_i|x)\left[-2\alpha(x - y_i)\right] - P(y_i|x)\sum_{j=1}^{L} P(y_j|x)\left[-2\alpha(x - y_j)\right] \tag{13}$$

$$= -2\alpha P(y_i|x)\left[(x - y_i) - \sum_{j=1}^{L} P(y_j|x)(x - y_j)\right] \tag{14}$$

$$= 2\alpha P(y_i|x)\left[y_i - \sum_{j=1}^{L} y_j P(y_j|x)\right], \tag{15}$$

$$\frac{\partial}{\partial x}\phi(x) = 2\alpha \sum_{i=1}^{L} z_i P(y_i|x)\left[y_i - \sum_{j=1}^{L} y_j P(y_j|x)\right] \tag{16}$$

$$= 2\alpha \left[\sum_{i=1}^{L} z_i y_i P(y_i|x) - \left(\sum_{i=1}^{L} z_i P(y_i|x)\right)\left(\sum_{j=1}^{L} y_j P(y_j|x)\right)\right] \tag{17}$$

$$= 2\alpha \cdot \text{Cov}(\mathbf{z}, \mathbf{y} \mid x). \tag{18}$$

Note that equation 18 implies that the partial derivative of $\phi(x)$ with respect to $x$ is equal to the covariance between $\boldsymbol{z}$ and $\boldsymbol{y}$ calculated according to $P(\cdot|x)$, which is nonzero unless $\boldsymbol{z}$ and $\boldsymbol{y}$ are uncorrelated. Since $\boldsymbol{z}$ generally depends on $\boldsymbol{y}$, this shows that $\phi(x)$ indeed does not have vanishing gradients.

In the special case when $\hat{\mathcal{A}} = \{0, \pm q, \pm 2q, \ldots, \pm kq\}$, equation 18 implies

$$\frac{\partial}{\partial x}\phi(x) = 2\alpha q\left[\sum_{i=-k}^{k} i z_i P(iq|x) - \left(\sum_{i=-k}^{k} z_i P(iq|x)\right)\left(\sum_{j=-k}^{k} j P(jq|x)\right)\right]. \tag{19}$$

For $\boldsymbol{y}$, we have

$$\frac{\partial}{\partial y_i} P(y_i|x) = P(y_i|x)\left[2\alpha(x - y_i)\right] - P^2(y_i|x)\left[2\alpha(x - y_i)\right], \tag{20}$$

$$\frac{\partial}{\partial y_j} P(y_i|x) = -P(y_i|x)P(y_j|x)\left[2\alpha(x - y_j)\right] \quad \forall j \neq i, \tag{21}$$

$$\frac{\partial}{\partial y_j}\phi(x) = \sum_{i=1}^{L} z_i \frac{\partial}{\partial y_j} P(y_i|x) \tag{22}$$

$$= z_j \cdot P(y_j|x)\left[2\alpha(x - y_j)\right] - 2\alpha(x - y_j)P(y_j|x)\sum_{i=1}^{L} z_i P(y_i|x) \tag{23}$$

$$= 2\alpha(x - y_j)\, P(y_j|x)\left[z_j - \sum_{i=1}^{L} z_i\, P(y_i|x)\right]. \tag{24}$$

In view of equation 24 and the chain rule, we have

$$\frac{\partial}{\partial q}\phi(x) = \sum_{j=-k}^{k} 2\alpha(x-jq)P(jq|x)\left[z_j - \sum_{i=-k}^{k} z_i P(iq|x)\right] \cdot j \tag{25}$$

$$= 2\alpha \sum_{j=-k}^{k} j(x-jq)P(jq|x)\left[z_j - \sum_{i=-k}^{k} z_i P(iq|x)\right] \tag{26}$$

when $\hat{\mathcal{A}} = \{0, \pm q, \pm 2q, \ldots, \pm kq\}$.

For $z$, we have

$$\frac{\partial\phi(x)}{\partial z_i} = P(y_i|x) \tag{27}$$

in the general case and

$$\frac{\partial\phi(x)}{\partial z_i} = P(iq|x) \tag{28}$$

in the uniform case.

For $\alpha$, we have

$$\frac{\partial}{\partial\alpha}P(y_i|x) = P(y_i|x)\left[-(x-y_i)^2\right] - P(y_i|x)\sum_{j=1}^{L} P(y_j|x)\left[-(x-y_j)^2\right] \tag{29}$$

$$= -P(y_i|x)\left[(x-y_i)^2 - \sum_{j=1}^{L}(x-y_j)^2 P(y_j|x)\right], \tag{30}$$

$$\frac{\partial}{\partial\alpha}\phi(x) = \sum_{i=1}^{L} z_i \frac{\partial}{\partial\alpha}P(y_i|x) \tag{31}$$

$$= -\left[\sum_{i=1}^{L} z_i(x-y_i)^2 P(y_i|x) - \left(\sum_{i=1}^{L} z_i P(y_i|x)\right)\left(\sum_{j=1}^{L}(x-y_j)^2 P(y_j|x)\right)\right] \tag{32}$$

$$= -\mathrm{Cov}\left(\mathrm{z}, (x-\mathrm{y})^2|x\right) \tag{33}$$

in the general case, and

$$\frac{\partial}{\partial\alpha}\phi(x) = -\left[\sum_{i=-k}^{k} z_i(x-iq)^2 P(iq|x) - \left(\sum_{i=-k}^{k} z_i P(iq|x)\right)\left(\sum_{j=-k}^{k}(x-jq)^2 P(jq|x)\right)\right] \tag{34}$$

when $\hat{\mathcal{A}} = \{0, \pm q, \pm 2q, \ldots, \pm kq\}$.

## A.2 PROOF OF THEOREMS

We begin with the proof of Theorem 1.

*Proof of Theorem 1*: Without loss of generality, we assume that $[a, b] = [-1, 1]$. Otherwise, translation and scaling can be applied to convert $[a, b]$ to $[-1, 1]$ so that $g(x)$ is defined over $[-1, 1]$. Since $g$ is continuous on the closed interval $[-1, 1]$, it is uniformly continuous. Let $\omega(\delta) = \sup\{|g(x) - g(y)| : |x - y| \le \delta, \ x, y \in [a, b]\}$ be the modulus of continuity. Then $\omega(\delta) \to 0$ as $\delta \downarrow 0$.

Choose $q > 0$ small enough so that

$$\omega(q) < \frac{\varepsilon}{3}.$$

Let $k = \lfloor 1/q \rfloor$. Consider

$$\hat{\mathcal{A}} = \{0, \pm q, \pm 2q, \ldots, \pm kq\}.$$

For each $-k \le i \le k$, let $z_i = g(iq)$, and

$$C_i = \begin{cases} [-1, (-k + \frac{1}{2})q] & \text{if } i = -k \\ ((i - \frac{1}{2})q, (i + \frac{1}{2})q] & \text{if } -k < i < k \\ (k - \frac{1}{2})q, 1] & \text{if } i = k \end{cases} . \tag{35}$$

With $\hat{\mathcal{A}}$ (hence $\boldsymbol{y}$) and $\boldsymbol{z}$ specified, we write the corresponding $\phi(x)$ as $\phi(x|q, \boldsymbol{z}, \alpha)$. Define a step function $f(x)$ as follows

$$f(x) = \sum_{i=-k}^{k} z_i \mathcal{X}_{C_i}(x). \tag{36}$$

Then it follows that for any $x$, and any $0 \le \lambda \le 1$,

$$\begin{aligned}
&|\phi(x|q, \boldsymbol{z}, \alpha) - f(x)| \\
&= |\phi(x|\boldsymbol{y}, \boldsymbol{z}, \alpha) - \lambda z_{i(x)} - (1 - \lambda)z_{j(x)} + (\lambda z_{i(x)} + (1 - \lambda)z_{j(x)} - f(x))| \\
&= |\phi(x|\boldsymbol{y}, \boldsymbol{z}, \alpha) - \lambda z_{i(x)} - (1 - \lambda)z_{j(x)} + \lambda(z_{i(x)} - f(x)) + (1 - \lambda)(z_{j(x)} - f(x))| \\
&\le |\phi(x|\boldsymbol{y}, \boldsymbol{z}, \alpha) - \lambda z_{i(x)} - (1 - \lambda)z_{j(x)}| + \omega(q). \tag{37}
\end{aligned}$$

In the above, equation 37 is due to the fact that in view of the definitions of $f(x), i(x), j(x)$, and $z_i$, $f(x)$ is equal to either $z_{i(x)}$ or $z_{j(x)}$, and $|z_{i(x)} - z_{j(x)}| \le \omega(q)$. Since equation 37 is valid for any $x$ and any $0 \le \lambda \le 1$, we have

$$|\phi(x|q, \boldsymbol{z}, \alpha) - f(x)| \le \min_{0 \le \lambda \le 1} |\phi(x|\boldsymbol{y}, \boldsymbol{z}, \alpha) - \lambda z_{i(x)} - (1 - \lambda)z_{j(x)}| + \omega(q) \tag{38}$$

for any $x$, which further implies that

$$\begin{aligned}
\sup_x |\phi(x|q, \boldsymbol{z}, \alpha) - f(x)| &\le \sup_x \min_{0 \le \lambda \le 1} |\phi(x|\boldsymbol{y}, \boldsymbol{z}, \alpha) - \lambda z_{i(x)} - (1 - \lambda)z_{j(x)}| + \omega(q) \\
&\le \frac{\varepsilon}{3} + \omega(q) \tag{39}
\end{aligned}$$

for large $\alpha$, where equation 39 follows from Property P2.

Finally,

$$\begin{aligned}
\sup_x |\phi(x|q, \boldsymbol{z}, \alpha) - g(x)| &= \sup_x |\phi(x|q, \boldsymbol{z}, \alpha) - f(x) + f(x) - g(x)| \\
&\le \sup_x |\phi(x|q, \boldsymbol{z}, \alpha) - f(x)| + \sup_x |f(x) - g(x)| \\
&\le \frac{\varepsilon}{3} + 2\omega(q) \\
&\le \varepsilon. \tag{40}
\end{aligned}$$

This completes the proof of Theorem 1.

We next prove Theorem 2.

*Proof of Theorem 2*: It suffices to prove Theorem 2 when $P_\alpha(\cdot|x)$ is the discrete conditional Gaussian distribution in Example 1. A similar argument applies to the discrete conditional Laplacian distribution.

Fix $\boldsymbol{y}$ and $\boldsymbol{z}$. Let

$$Z = \max\{|z_i| : 1 \le i \le L\} \text{ and } s = \min\{|y_i - y_j| : 1 \le i, j \le L, i \ne j\}.$$

For each $x$, let $y_{i^*(x)}$ be the nearest point of $x$ in $\hat{\mathcal{A}}$. Note that $i^*(x)$ is equal to either $i(x)$ or $j(x)$, and for any $i \notin \{i(x), j(x)\}$,

$$|x - y_i| - |x - y_{i^*(x)}|$$

is at least a positive integer multiplier of $s$. Then we have

$$
\begin{aligned}
|\phi(x) &- z_{i(x)} P_\alpha(y_{i(x)}|x) - z_{j(x)} P_\alpha(y_{j(x)}|x)| \\
&= |\sum_{i \neq i(x), j(x)} z_i P_\alpha(y_i|x)| \\
&\leq Z \sum_{i \neq i(x), j(x)} P_\alpha(y_i|x) \\
&\leq Z \sum_{i \neq i(x), j(x)} \frac{e^{-\alpha(x-y_i)^2}}{e^{-\alpha(x-y_{i^*(x)})^2}} \\
&\leq 2Z \sum_{j=1}^{\infty} e^{-\alpha s^2 j^2} \\
&\leq 2Z \sum_{j=1}^{\infty} e^{-\alpha s^2 j} \\
&= 2Z \frac{e^{-\alpha s^2}}{1 - e^{-\alpha s^2}}
\end{aligned}
\tag{41}
$$

Let

$$
r = \sum_{i \neq i(x), j(x)} P_\alpha(y_i|x).
$$

In view of equation 41 and its derivation, it follows that

$$
\begin{aligned}
\min_{0 \leq \lambda \leq 1} &|\phi(x) - \lambda z_{i(x)} - (1-\lambda) z_{j(x)}| \\
&\leq |\phi(x) - z_{i(x)}(P_\alpha(y_{i(x)}|x) + r/2) - z_{j(x)}(P_\alpha(y_{j(x)}|x) + r/2)| \\
&\leq |\phi(x) - z_{i(x)} P_\alpha(y_{i(x)}|x) - z_{j(x)} P_\alpha(y_{j(x)}|x)| + rZ \\
&\leq 2Z \frac{e^{-\alpha s^2}}{1 - e^{-\alpha s^2}} + rZ \\
&\leq 4Z \frac{e^{-\alpha s^2}}{1 - e^{-\alpha s^2}}.
\end{aligned}
\tag{42}
$$

Note that the upper bound in equation 42 is independent of $x$. Letting $\alpha$ go to $\infty$, Theorem 2 follows. This completes the proof of Theorem 2.

### A.3 IMPLEMENTATION DETAILS

#### A.3.1 TRAINING RECIPES

**Function fitting.** We train a model for 40K iterations, with a batch size of 98, using Adam optimizer without weight decay, a learning rate of 0.001 for the 1D function and 0.01 for the 2D function. The loss function is MSE. More details can be found in Chelly et al. (2024) and their released source code.

**Image fitting.** The target images are resized to $128 \times 128$, and pixel coordinates are rescaled to be within $[-1, 1]^2$. We train a model with batch size 1024 for 1000 epochs, i.e. 16K iterations, using Adam optimizer without weight decay. The initial learning rate is 0.001 and decays by 0.1 at the 500th epoch. The loss function is MSE.

**Toy classification.** We train a model for 100 epochs with batch gradient descent using Adam optimizer with a learning rate of 0.001 and without weight decay. The loss function is the cross-entropy loss.

**CNNs on CIFAR-100.** We train for 240 epochs with an initial learning rate of 0.05 by default, which decays by 0.1 at the 150th, 180th, and 210th epochs. For ShuffleNetV1, a smaller initial learning rate of 0.01 is used. Batch size is set to 64, and the optimizer is SGD. More details, including KD

Table 8: The training recipe for transformer-based models on CIFAR-100.

| Setting | Configuration |
|---|---|
| training epochs | 300 |
| batch size | 512 |
| optimizer | AdamW |
| base learning rate | 2e-3 |
| weight decay | 0.05 |
| optimizer momentum | $\beta_1, \beta_2 = 0.9, 0.999$ |
| learning rate decay | cosine |
| warmup epochs | 50 |
| warmup schedule | linear |
| layer-wise lr decay (Clark et al., 2020; Bao et al., 2022) | None |
| randaugment (Cubuk et al., 2020) | (9, 0.5) |
| mixup (Zhang et al., 2017) | 0.8 |
| cutmix (Yun et al., 2019) | 1.0 |
| random erasing (Zhong et al., 2020) | 0.25 |
| label smoothing (Szegedy et al., 2016) | 0.1 |
| layer scale (Touvron et al., 2021b) | 1e-6 |
| head init scale (Touvron et al., 2021b) | None |
| gradient clip | None |

settings, can be found in the source code released by Tian et al. (2019). Note that the distillation method used in our experiments is just the default KD as in Hinton et al. (2015).

**Transformer-based models on CIFAR-100.** The settings are summarized in Tab. 8.

**ResNets and Swin-T on ImageNet-1K.** We follow the recipe given on the PyTorch official website (maintainers & contributors, 2016).

**ViT-T on ImageNet-1K.** We follow the configuration specified in Tab. 8, except that batch size is now 4096.

**LLM finetuning.** The settings are summarized in Tab. 9.

Table 9: The training recipe for GPT-2 fine-tuning.

| Setting | Configuration |
|---|---|
| Batch Size | 32 |
| Optimizer | RMSprop |
| Learning Rate | $5 \times 10^{-7}$ with linear warmup steps of 150 |
| Trainer | FSDPTrainer |
| Max Gradient Norm | 10.0 |
| Max Length for an Input (Prompt + Response) | 512 |
| Max Length for Prompt | 256 |

### A.3.2 SQUAF-RELATED SETTINGS

**Function fitting, image fitting and toy classification.** For SQUAF within MLPs, we set $k = 2$, initialize $q$ with 0.5 and $\alpha$ with 5, and each $z_i$ is initialized by sampling from a uniform distribution over $[-1, 1]$. For SQUAF-only experiments in function fitting, we set $k = 95$ or $k = 13$ for two cases in Tab. 1, initialize $q$ with $1/k$ so that $\hat{\mathcal{A}}$ covers the domain $[-1, 1]$ and freeze it during training (thus not counted into trainable parameters), initialize $\alpha = 1.5/q^2$ for a relatively smooth SQUAF shape, and initialize each $z_i$ by sampling from a uniform distribution over $[-1, 1]$.

**CIFAR-100.** We first fit SQUAFs to ReLU and GELU as how we conducted the SQUAF-only 1D function fitting experiments, except that $z_i$ is now initialized with $iq$ for $i \geq 0$ and 0 for $i < 0$. These SQUAF-fitted-ReLU and SQUAF-fitted-GELU serve as SQUAF initialization in CNNs and transformer-based networks, respectively. We select $k = 16$, initial $q = 1$ for CNNs while initial

$q = 0.25$ for transformer-based networks, since we observe that the pre-activations in transformer-based networks have a smaller variance at the time of initialization. As for SQUAF-P, $k = 16$, initial $q$ is set to a smaller value $0.125$ for VGG-8 to control quantization error, while initial $q$ remains $0.25$ for ViT-T. For all experiments on CIFAR-100, we only train the SQUAF parameters in the first SQUAF, i.e., $(q_1, \boldsymbol{z}_1, \alpha_1)$ while freezing $\{(q_t, \boldsymbol{z}_t, \alpha_t)\}_{t=2}^{T-1}$, since we find that training all SQUAFs tends to cause overfitting as they inject more nonlinearity into the network than needed for this dataset. Also, note that SQUAF parameters are always exempted from weight decay. In the case of CNNs, SQUAFs are trained by an Adam optimizer with learning rate $0.001$, which is different from the SGD optimizer used for other model weights $W$. To make a fair comparison, parameters in other trainable AFs are also exempted from weight decay, and also use the above separate optimizer in CNN cases.

**ImageNet-1K.** The initialization of SQUAF is the same as CIFAR-100, i.e., SQUAF-fitted-ReLU for CNNs and SQUAF-fitted-GELU for transformer-based networks, except that $q = 1$ for all four models. For DNNs with stages, i.e., ResNets and Swin-T, SQUAFs in the last stage are frozen to prevent overfitting, while all SQUAFs in ViT-T are trained. As before, SQUAF parameters are exempted from weight decay.

**LLM fine-tuning.** We use the same SQUAF-fitted-GELU as before with $q = 1$ and $k = 16$ as initialization, and all SQUAFs are trained in both SFT and DPO stages.

## A.4 COMPUTATIONAL EFFICIENCY

Tab. 10 shows the inference latency and training time of SQUAF and SQUAF-P compared to other AFs for ViT-T on CIFAR-100. Note that for each $x$, the support of CPMF $P(\cdot|x)$ is constrained to the 5 nearest points of $x$ in $\hat{\mathcal{A}}$, instead of the entire $\hat{\mathcal{A}}$, to reduce complexity, as suggested in Yang & Hamidi (2025) and Salamah et al. (2025), and this is applied to all our experiments in this paper. As shown in Tab. 10, SQUAF is faster than other complex AFs such as PolyNorm and DiTAC, while SQUAF-P is slightly less computationally efficient than SQUAF due to the need of sampling.

Table 10: Comparison of computational efficiency. The inference latency is evaluated on an NVIDIA RTX A6000 GPU, and the training time is measured for distributed training on 8 NVIDIA RTX A6000 GPUs.

| AF | Inference Latency (ms/img) | Training Time (s/epoch) |
|---|---|---|
| ReLU | 5.727 | 9 |
| GELU | 5.896 | 9 |
| LReLU | 5.792 | 9 |
| PReLU | 5.834 | 9 |
| Swish | 6.185 | 9 |
| CRReLU | 6.959 | 10 |
| PolyReLU | 7.241 | 11 |
| PolyNorm | 8.694 | 12 |
| DiTAC | 9.642 | 19 |
| SQUAF | 8.399 | 11 |
| SQUAF-P | 8.930 | 12 |
| SQUAF-S | 7.022 | n/a |

To accelerate SQUAF-P during inference, let us introduce SQUAF-S, another variant of SQUAF, denoted as $\phi_s(x)$, which is essentially a step function. Particularly,

$$\phi_s(x|\boldsymbol{y}, \boldsymbol{z}) = \sum_{i=1}^{L} z_i \mathcal{X}_{C_i}(x), \tag{43}$$

where $C_1 = (-\infty, \frac{y_1+y_2}{2})$, $C_i = [\frac{y_{i-1}+y_i}{2}, \frac{y_i+y_{i+1}}{2})$ for $i = 2, \ldots, L-1$, and $C_L = [\frac{y_{L-1}+y_L}{2}, +\infty)$. In the uniform case, this is simply

$$\phi_s(x|q, \boldsymbol{z}) = z_{\min\{\max\{\lceil x/q \rceil, -k\}, k\}}, \tag{44}$$

where $\lceil \cdot \rceil$ denotes the rounding operation.

Replacing SQUAF-P $\phi_p(x|\boldsymbol{y}, \boldsymbol{z}, \alpha)$ (or $\phi_p(x|q, \boldsymbol{z}, \alpha)$) with SQUAF-S $\phi_s(x|\boldsymbol{y}, \boldsymbol{z})$ (or $\phi_s(x|q, \boldsymbol{z})$) after training can result in reduced latency (comparable to some simple AFs such as CRReLU, see Tab. 10), deterministic output and won't degrade accuracy performance much as long as the learned $\alpha$ is adequately large. For example, we incorporate SQUAF-Ps into ViT-T and train it on CIFAR-100 following equation 11 with $\lambda = 0$. The trained model reaches 72.24% Top-1 validation accuracy which is higher than all baseline AFs compared in Tab. 6. For this model, there's actually no accuracy degradation when we switch from SQUAF-P to SQUAF-S in post-training inference, i.e., 72.24% accuracy maintained.

## A.5 MORE RESULTS ON LLMS

Using DPO, we fine-tune GPT-2 XL (1.5B parameters) and Pythia-2.8B (2.8B parameters) on Anthropic HH dataset following the standard recipe given by Rafailov et al. (2023). The results are shown in Tab. 11, and it's verified that SQUAF can obtain consistent gains on larger LLMs.

Table 11: DPO results on larger LLMs. $\beta$ is set to 0.1 following Rafailov et al. (2023).

| | | ERA (%) ↑ | ERM ↑ | EL ↓ |
|---|---|---|---|---|
| GPT-2 XL (1.5B) | GELU | 62.89 | 0.2182 | 0.6411 |
| | SQUAF | **63.28** (+0.39) | **0.2190** | **0.6400** |
| Pythia-2.8B | GELU | 63.67 | 0.3883 | 0.6108 |
| | SQUAF | **64.84** (+1.17) | **0.4302** | **0.6075** |

## A.6 ANALYSIS ON INITIALIZATION OF $q$ AND $\alpha$ FOR IMAGE FITTING

We've conducted a grid search over initial $q$ and $\alpha$ on the Camera image fitting task to analyze how sensitive the DNN performance is w.r.t. the initialization of $q$ and $\alpha$. Note that across all four SQUAFs in the MLP, the same $q$ and $\alpha$ is used for initialization. To determine the proper searching range of $q$, we record the mean and standard deviation (std) of the first batch of pre-activations at each of the four layers, where ReLU is temporarily used as the AF in the MLP. Results show that pre-activations are roughly zero-mean for each layer, and the standard deviations are 0.51, 0.20, 0.10 and 0.06 for four layers, respectively. So, we decide to search $q$ within $[0.1, 1]$. As for $\alpha$, we set it by tuning the gradient scaling constant $\hbar = \alpha q^2$ (Salamah et al., 2025) which controls the smoothness of SQUAFs. Tab. 12 shows the PSNR results obtained with different $(q, \hbar)$ pairs, and we can conclude that as long as $q$ is not too large compared to the scale of pre-activations and $\hbar$ is not too large to maintain relatively smooth SQUAF shapes, then SQUAF always outperforms all compared AFs in Tab. 1 with a large margin.

Table 12: PSNR results for fitting the Camera image with different initialization of $q$ and $\alpha$. ↑ indicates the performance is better than all compared AFs in Tab. 1 — the best baseline AF achieves 31.76 dB PSNR — while ↓ indicates the performance is worse than some of the baseline AFs.

| PSNR (dB) | $\alpha = 0.5/q^2$ | $\alpha = 1/q^2$ | $\alpha = 2/q^2$ | $\alpha = 5/q^2$ | $\alpha = 10/q^2$ |
|---|---|---|---|---|---|
| $q = 0.1$ | 44.90 (↑) | 51.60 (↑) | 52.89 (↑) | 46.73 (↑) | 24.09 (↓) |
| $q = 0.2$ | 43.08 (↑) | 47.09 (↑) | 53.75 (↑) | 54.74 (↑) | 37.18 (↑) |
| $q = 0.5$ | 40.33 (↑) | 44.35 (↑) | 42.40 (↑) | 45.57 (↑) | 41.99 (↑) |
| $q = 1.0$ | 30.86 (↓) | 37.00 (↑) | 36.67 (↑) | 42.13 (↑) | 39.37 (↑) |

## A.7 NUMBER OF TRAINABLE PARAMETERS IN SQUAFS

In Tab.13, we list the number of trainable parameters for each DNN used in image classification and LLM finetuning before incorporating SQUAFs, and the number of trainable parameters in SQUAFs corresponding to that DNN. Clearly, the number of additional trainable parameters introduced by SQUAFs is negligible compared to the original weights in DNNs, leading to an increase rate of at most 0.0076%.

Table 13: The number of trainable parameters in original DNNs compared to the number of additional trainable parameters introduced by SQUAFs.

| | Model | ShuffleNetV1 | VGG-8 | ResNet-56 | VGG-13 | ViT-T | EfficientFormer-L1 |
|---|---|---|---|---|---|---|---|
| CIFAR-100 | Original Trainable Param. | 949258 | 3965028 | 856420 | 9462180 | 5543716 | 11481728 |
| | SQUAF Trainable Param. | 35 | 35 | 35 | 35 | 35 | 35 |
| | Increase Rate | 0.00369% | 0.00088% | 0.00409% | 0.00037% | 0.00063% | 0.00030% |
| | Model | ResNet-18 | ResNet-34 | ViT-T | Swin-T | | |
| ImageNet-1K | Original Trainable Param. | 11689512 | 21797672 | 5543716 | 28288354 | | |
| | SQUAF Trainable Param. | 455 | 945 | 420 | 350 | | |
| | Increase Rate | 0.0039% | 0.0043% | 0.0076% | 0.0012% | | |
| | Model | GPT-2 | | | | | |
| LLM FT | Original Trainable Param. | 124439808 | | | | | |
| | SQUAF Trainable Param. | 420 | | | | | |
| | Increase Rate | 0.00034% | | | | | |

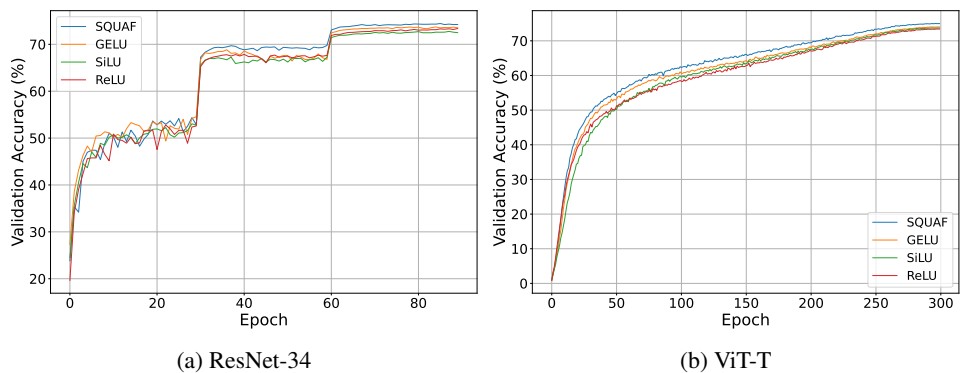

(a) ResNet-34              (b) ViT-T

Figure 4: Convergence of validation accuracy on ImageNet-1K.

## A.8 CONVERGENCE STABILITY

In Fig. 4, we plot the convergence curves of validation accuracy for ResNet-34 and ViT-T trained on ImageNet-1K, including those for SQUAF and compared AFs. It's shown that the convergence of SQUAF is as stable as other well-established AFs, while achieving better final accuracy performance.

## A.9 GENERALIZATION GAP FOR KD

In Fig. 5, we plot the curves for generalization gap, i.e., training accuracy minus validation accuracy, across epochs for both students used in Tab. 7, including those for SQUAFs and baseline AFs. Note that ReLU is not included as we cite its results from the literature (Tian et al., 2019) without rerunning those experiments, and thus no training dynamics are recorded; also, PolyReLU is not included as it doesn't converge. As shown in Fig. 5, SQUAF achieves the lowest generalization gap among all compared AFs at the end of training, which means that it alleviates overfitting compared to baseline AFs. Additionally, the generalization gap comparison is also numerically reported in Tab. 14, further validating our claim quantitatively.

## A.10 VISUALIZATION OF SQUAFs

We visualize the initial and learned SQUAFs for ResNet-18, ResNet-34, ViT-T, and Swin-T on ImageNet-1K, shown in Figures 6, 7, 8, and 9, respectively. Interestingly, a dent near the origin is a very common pattern in the learned SQUAFs, which echoes the designs of SiLU and GELU where that dent is incorporated to the canonical ReLU.

## A.11 LLM USAGE

In this paper, LLMs are used to find related work and polish writing.

Table 14: Comparison of generalization gaps resulting from different AFs at the end of KD training. Results are averaged over 3 runs.

| AF | Generalization Gap (%) ↓ | |
| | VGG-13 → VGG-8 | WRN-40-2 → WRN-16-2 |
| --- | --- | --- |
| GELU | 22.57 | 14.87 |
| LReLU | 21.52 | 13.76 |
| PReLU | 23.43 | 14.31 |
| Swish | 22.52 | 14.95 |
| CRReLU | 24.78 | 15.61 |
| PolyNorm | 24.23 | 16.50 |
| DiTAC | 22.53 | 15.54 |
| SQUAF | 20.68 | 13.12 |

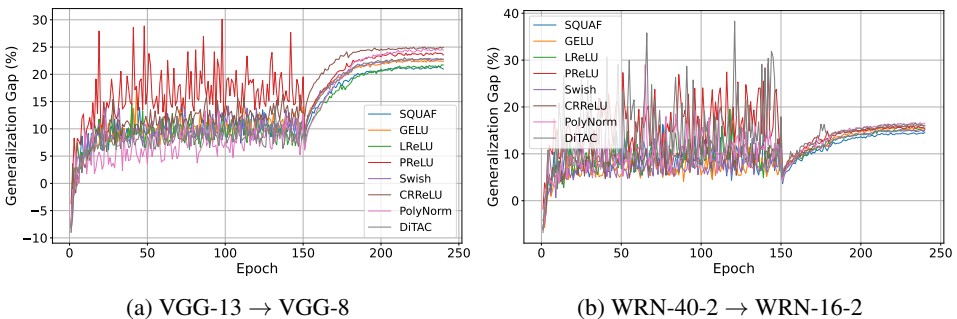

(a) VGG-13 → VGG-8

(b) WRN-40-2 → WRN-16-2

Figure 5: Generalization gap across epochs for students in Tab. 7.

## A.12 KD ON IMAGENET-1K

To further demonstrate SQUAF's effectiveness in KD, we conduct additional experiments on ImageNet-1K by distilling a ResNet-18 teacher into a MobileNet student (Howard et al., 2017), following the standard distillation setup in Tian et al. (2019). Table 15 reports the Top-1 accuracy and number of trainable parameters for the teacher, the ReLU-based student, and the SQUAF-based student. The results show that a MobileNet equipped with SQUAF not only achieves a substantial improvement over the ReLU counterpart, but also outperforms the ResNet-18 teacher, despite using only about one-third of the teacher's parameters. In contrast, the MobileNet student using ReLU does not surpass the teacher. These findings provide additional strong evidence that SQUAF significantly enhances the effectiveness of knowledge distillation on large-scale datasets.

Table 15: Top-1 validation accuracy (%) on ImageNet-1K and the number of trainable parameters in a model.

| Teacher | ResNet-18 | |
| | 69.76 | 11689512 |
| --- | --- | --- |
| Student | MobileNet | |
| ReLU | 69.71 | 4231976 |
| SQUAF | **70.87** (+1.16) | 4232781 |

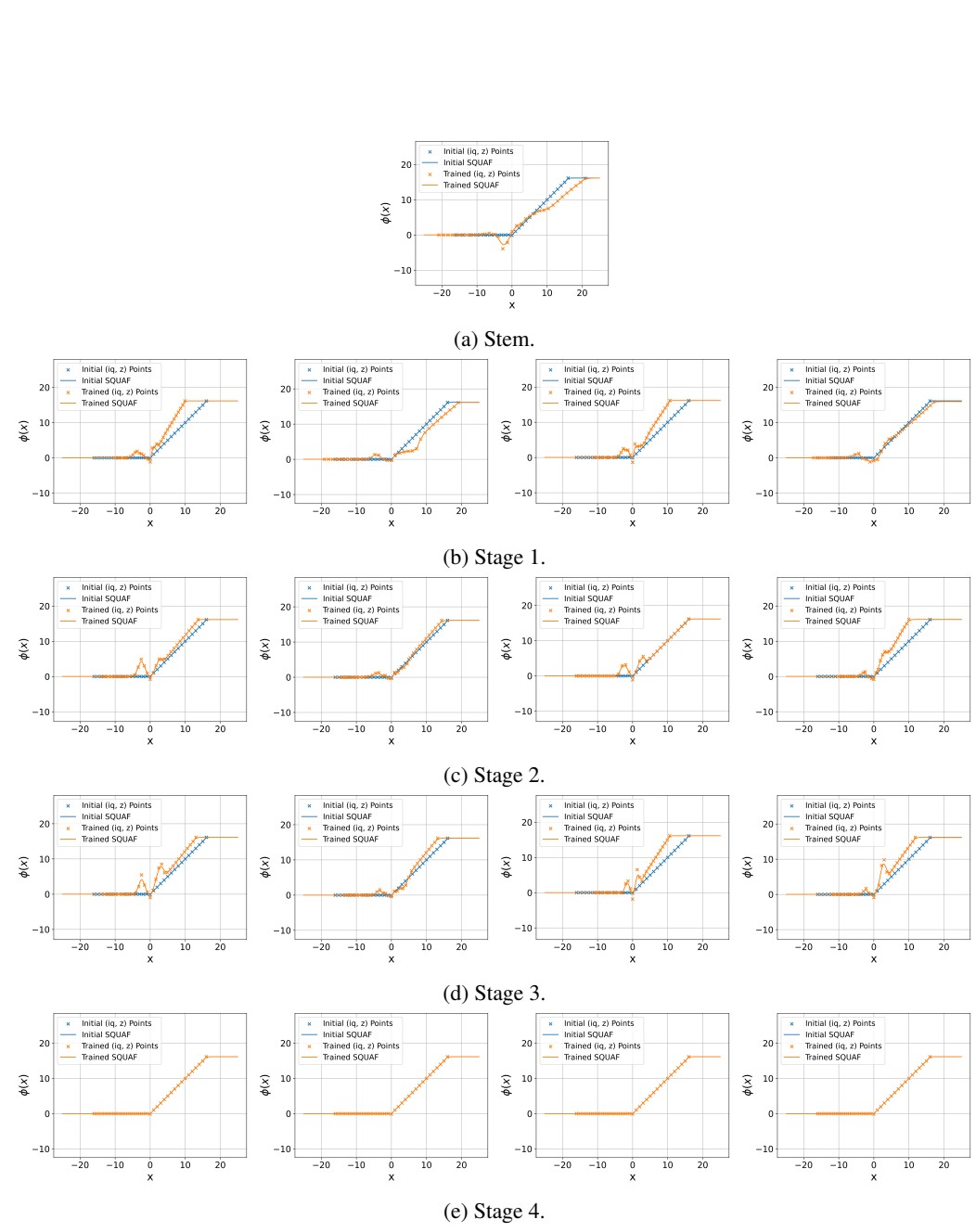

Figure 6: Visualization of SQUAFs in ResNet-18. The plots are arranged in raster order (top-to-bottom, left-to-right) corresponding to increasing depth in the network.

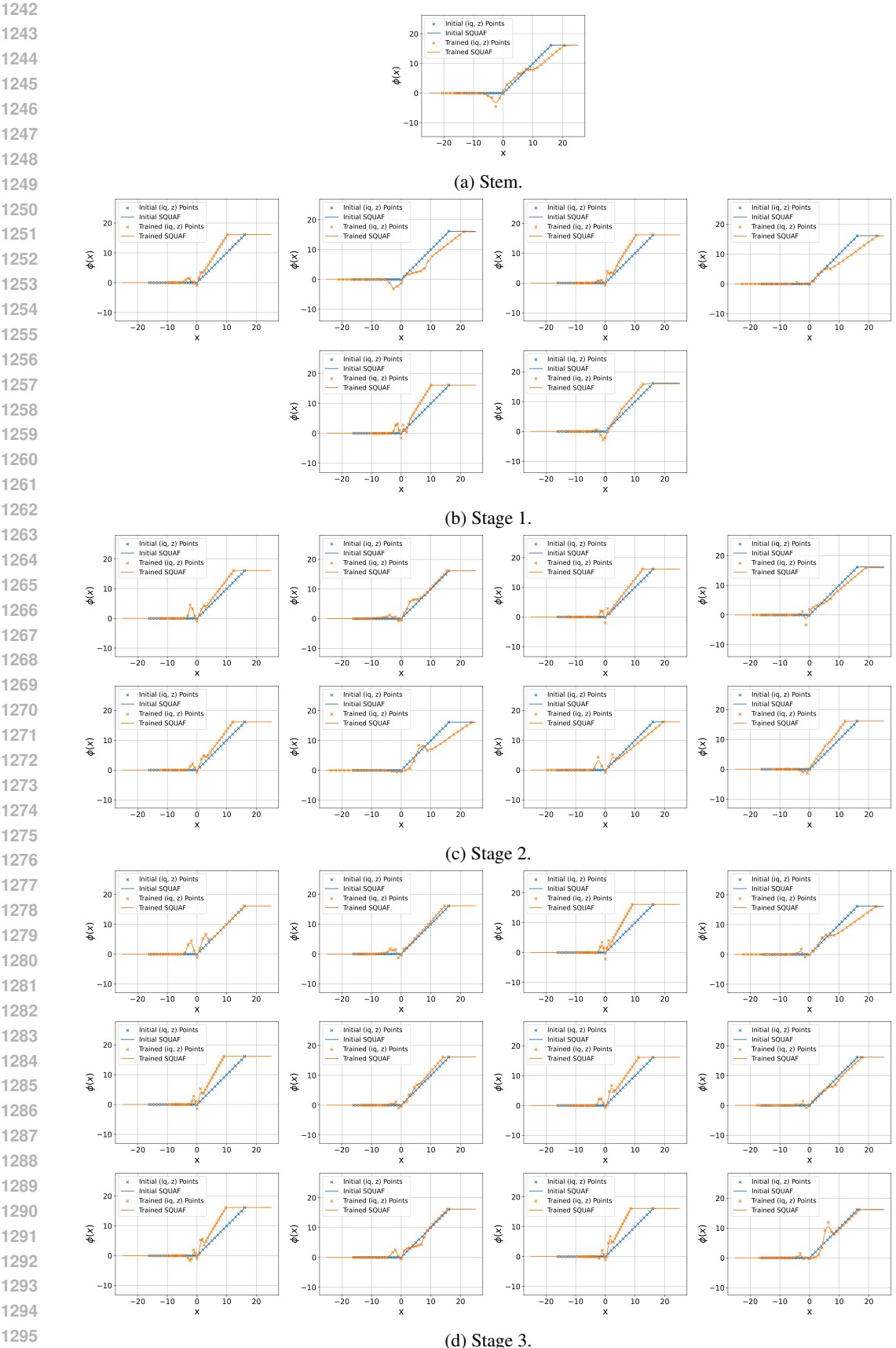

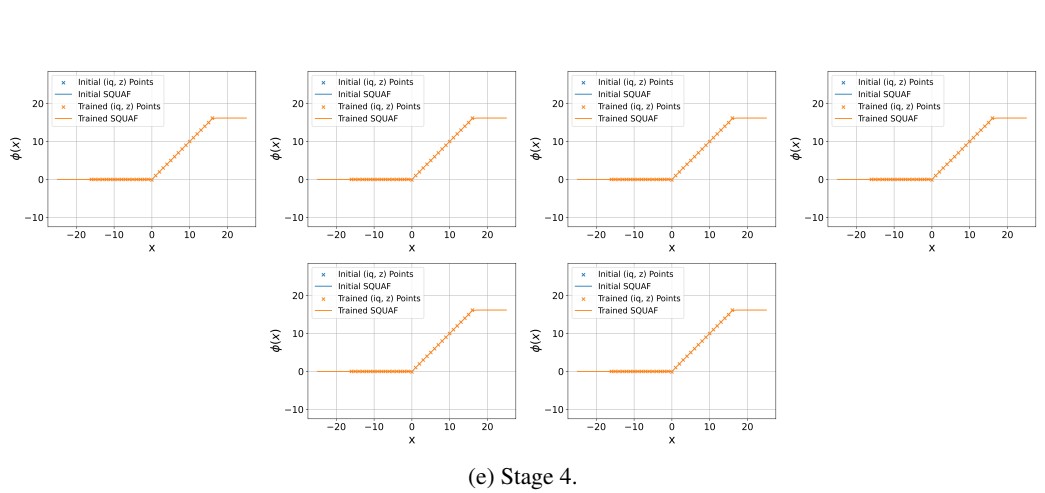

(e) Stage 4.

Figure 7: Visualization of SQUAFs in ResNet-34.

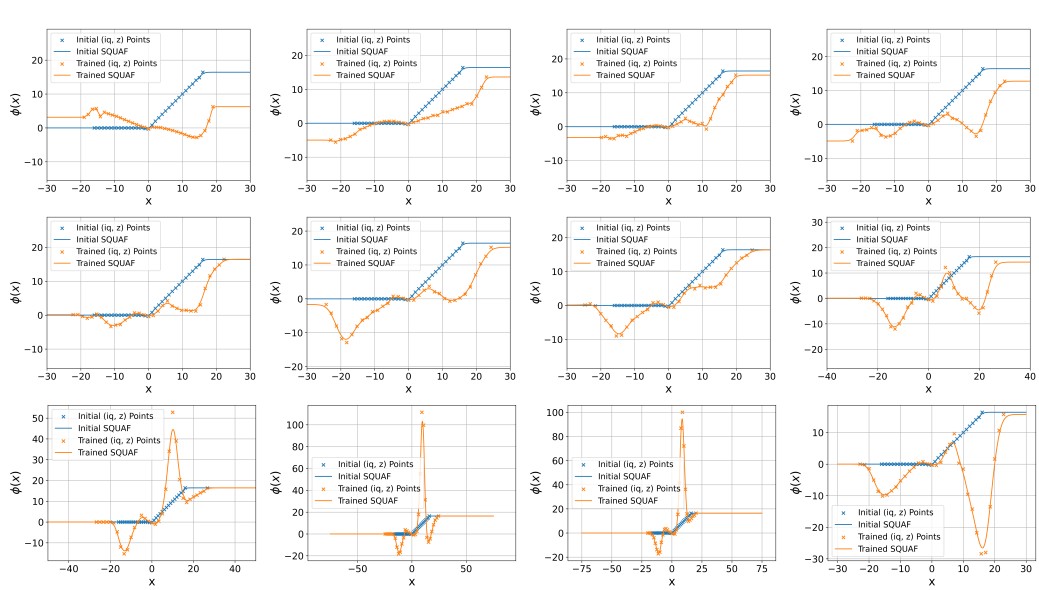

Figure 8: Visualization of SQUAFs in ViT-T.

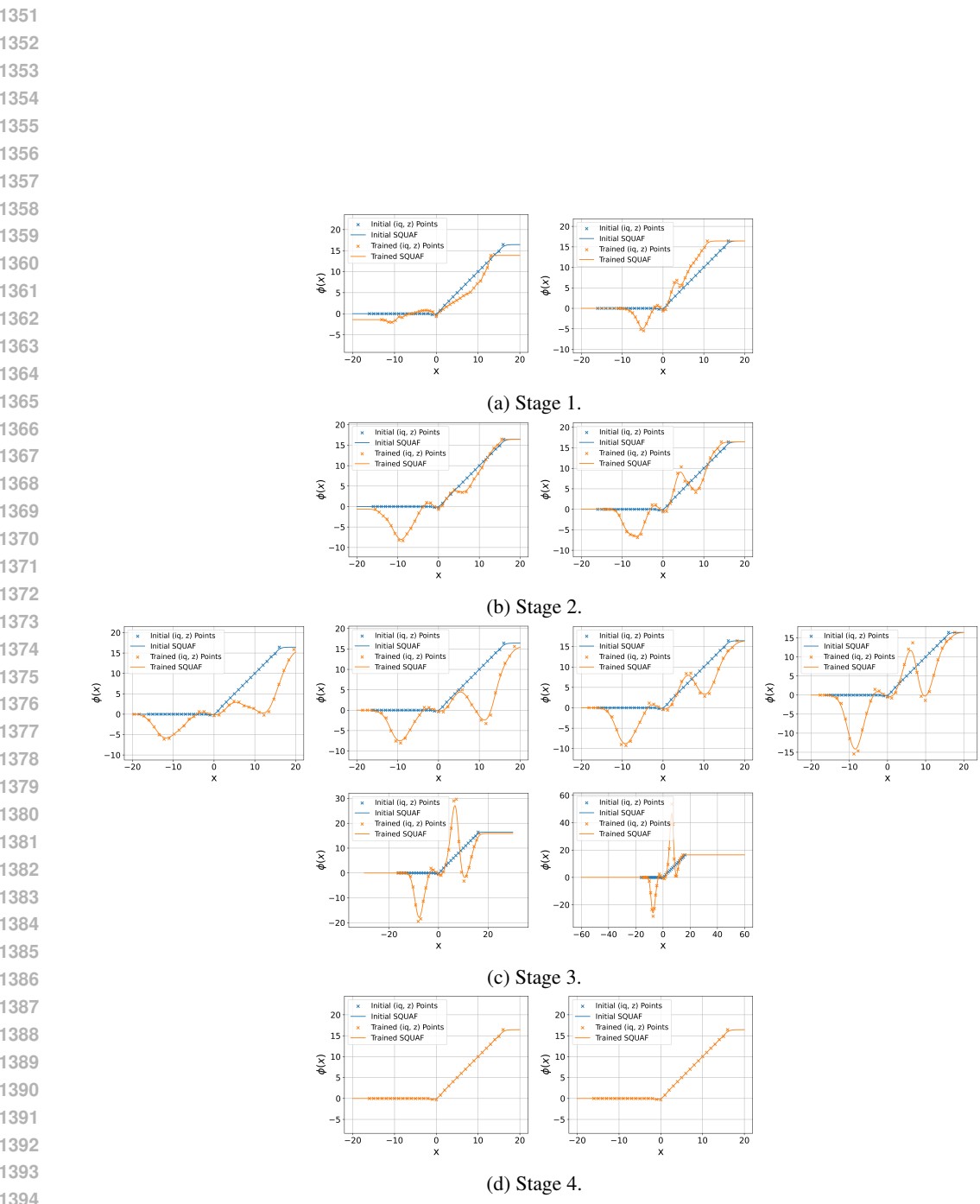

(a) Stage 1.

(b) Stage 2.

(c) Stage 3.

(d) Stage 4.

Figure 9: Visualization of SQUAFs in Swin-T.

