# OpenReview forum: "Soft Quantization Activation Functions For Deep Learning"
_ICLR.cc/2026/Conference — ICLR 2026 Conference Withdrawn Submission_

### Official Review · Reviewer_qi4Q · 2025-10-26

**Soundness:** 3
**Presentation:** 3
**Contribution:** 2
**Rating:** 4
**Confidence:** 5

**Summary:**

The proposed activation function (AF) is an impressive work highlighting the issues with simple AF such as ReLU which needs deeper models to capture non linearity and trainable AFs like PReLU, swish etc which can capture non linearity more effectively but by adding extra parameter makes distributed training more complicated (communication between different nodes to share updated parameters). Trainable, architecture-agnostic AF, Soft Quantization Activation Functions (SQUAFs) proposed by Authors can approximate any continuous nonlinear one-dimensional function with arbitrary precision. SQUAFs models are claimed to be better than counterpart bigger models trained using existing AFs. Additionally, it is shown to save more than 39x communication cost compared to models using trainable parameters in distributed computing setting. Results of four different tasks shows consistent improvement in performance relavie the the baseline models.

This paper targets an important issue and the formulation of the problem and solution is interesting but the improvement in accuracy is not significant (given the the impact of new AFs in training is also now known) and limitation of 1D approximation may limit model expressivity. Some of the good results are on very small datasets which does not convey the real impact of this method. This is a weak reject and the score can be changed based on Author’s explanation as indicated in the limitations section.

**Strengths:**

- Authors have proposed a new Activation function which is trainable (but with partial derivatives),  adaptable to quantization and it is architecture agnostic to improve the performance of four different class of models
- Mathematical formulation of SQUAF is interesting and intuitive and it focuses on overcoming the limitations of existing AF.
- SQUAF models consistently have high accuracy compared to baseline models
- Designed as a plug-and-play replacement for existing activations in any architecture.
- Detailed results comparing with other activation functions such as GELU, PRELY, SWISH and few more.

**Weaknesses:**

- Table 4 : ResNet18 baseline performance is using ReLU (69.76) but if you replace ReLU with slightly more complex AF (SiLU), it will improve accuracy to almost similar to what can be achieved by SQUAF. Authors are encouraged to check those results and discuss the merit of using SQUAF in this context.
- Authors are also encouraged to check the same for models trained on CIFAR as it can be trained much faster compared to ImageNet.
- Table 7 results are impressive but CIFAR100 is too small and non complex dataset so showing a competitive results on larger datasets will make their claim stronger.
- Overall the improvement in accuracy does not seem to be significantly high.
- Trainable activation parameters could introduce extra computational or tuning cost, especially in large models. it will also be interesting to see the impact of these AFs on training complexity/training time etc.
- It’s not clear how much gain comes purely from adaptability versus other design factors.
- The approximation proof is restricted to one-dimensional functions and generalization to high-dimensional activations may be non-trivial. This can result in expressivity gap. Without extending the theory, it’s unclear if SQUAF’s adaptivity at the neuron level translates into provably higher expressivity at the layer or network level.

**Questions:**

- Do Authors think that for resnet18/imagenet results achieved by SQUAF will be better compared to just replacing ReLU with SiLU
- What are the training overheads? can it be quantified?

---

> ### Author Response · Authors · 2025-11-25
> **Response to Reviewer qi4Q (Part 1)**
>
> We thank you very much for taking time to review our paper and provide valuable feedbacks. Below, please find our point-by-point responses to your comments.
>
> > Weakness 1 & 2: "Table 4 : ResNet18 baseline performance is using ReLU (69.76) but if you replace ReLU with slightly more complex AF (SiLU), it will improve accuracy to almost similar to what can be achieved by SQUAF. Authors are encouraged to check those results and discuss the merit of using SQUAF in this context." "Authors are also encouraged to check the same for models trained on CIFAR as it can be trained much faster compared to ImageNet."
>
> **Response**: In Table 4 of the revised manuscript (also shown in the table below), we extend our ImageNet-1K results by including three AFs, i.e., ReLU, GELU and SiLU, into the comparsion.
>
> ||ResNet-18|ResNet-34|ViT-T|Swin-T|
> |:---:|:---:|:---:|:---:|:---:|
> |Baseline|69.76|73.31|73.90|81.28|
> |ReLU|69.76 (+0.00)|73.31 (+0.00)|73.43 (-0.47)|80.70 (-0.58)|
> |GELU |70.66 (+0.90)|73.71 (+0.40)|73.90 (+0.00)|81.28 (+0.00)|
> |SiLU |70.51 (+0.75)|72.76 (-0.55)|73.70 (-0.20)|80.29 (-0.99)|
> |**SQUAF** |70.88 (+1.12)|74.44 (+1.13)|75.00 (+1.10)|81.77 (+0.49)|
>
> It can be seen that replacing ReLU with SiLU in ResNet-18 can indeed obtain a non-trivial gain in accuracy. However, it’s still 0.37% lower than that obtained by SQUAF. More importantly, for other three evaluated DNNs, SiLU degrades the accuracy significantly. In contrast, SQUAF provides consistent gains across all four architectures, including CNNs and Transformers. Note that GELU turns out to be a more competitive AF than SiLU, while still being significantly worse than SQUAF.
>
> As suggested, we also added SiLU to the CIFAR-100 comparisons in Table 6 of the revised manuscript. Again, SiLU performs poorly, degrading the accuracy for three out of 6 models and achieving a meaningful accuracy gain for only one out of 6 models. In contrast, SQUAF provides consistently significant gains across all 6 models, achieving the best in each case.
>
> > Weakness 3: Table 7 results are impressive but CIFAR100 is too small and non complex dataset so showing a competitive results on larger datasets will make their claim stronger.
>
> **Response**: Thank you for your positive comments on the Table 7 results. We would like to clarify the intended role of the CIFAR-100 KD experiments in our paper.
>
> Our work is not focused on developing a new knowledge distillation (KD) method; rather, the KD experiments serve to demonstrate the adaptability of SQUAFs across different training paradigms, including KD. Since CIFAR-100 is a widely used benchmark for KD evaluations, it provides a standard and well-understood setting in which to isolate and study the effect of replacing the activation function.
>
> Importantly, the strength of SQUAFs on large-scale and complex datasets is shown extensively in other parts of the paper. Sections 4.2 and 4.3 and Appendix A.5 include large-scale image classification and LLM experiments, and—as discussed in our response to Weakness 2 raised by Reviewer 1Xpj—SQUAFs achieve strong gains on LLMs of realistic scales (1B–8B). These results collectively demonstrate that SQUAFs are effective not only on CIFAR-100 but also on substantially larger and more complex tasks.

---

> > ### Comment · Reviewer_qi4Q · 2025-11-27
> > **Thank you for updated comments.**
> >
> > Thank you for updated comments and revised version of the paper. Lower accuracy for some models with SilU compared to ReLU does not seem to be intuitive as most of the models either perform better or at least at same level with SiLU. Authors are advised to look at following papers
> > * [SEARCHING FOR ACTIVATION FUNCTIONS](https://arxiv.org/pdf/1710.05941)
> > * [ActNAS : Generating Efficient YOLO Models using Activation NAS](https://arxiv.org/html/2410.10887v1)
> > However, I agree with the benefits on other models as suggested by Authors.

---

> > > ### Author Response · Authors · 2025-11-27
> > >
> > > Thank you for pointing us to these two references. We have carefully reviewed both papers and found that they do not contradict our results.
> > >
> > > In the Swish/SiLU paper [1], the authors conduct **per-activation hyperparameter tuning** and explicitly note: _“For each activation function, we try 3 different learning rates with RMSProp and pick the best”_ in their Section 5.3. This means that SiLU’s reported gains are **not plug-and-play**, but rely on activation-specific tuning. In contrast, our comparisons use the **same default training recipe** for all activation functions, with no per-activation tuning. Under such settings, SiLU is not guaranteed to outperform ReLU (or GELU), since the default training recipes in CNNs and transformers are typically optimized for **ReLU** and **GELU**, respectively.
> > >
> > > Similarly, in the ActNAS paper [2], the default AF in YOLO5n/YOLO8m is SiLU, so the model design and training recipe are naturally **tuned for SiLU**. The observed strong performance of SiLU in that context could be (partially) attributed to this inherent bias, and does not imply that SiLU will generalize well across different architectures, training recipes, datasets, or tasks.
> > >
> > > In fact, several other papers also report that Swish/SiLU is not consistently superior to ReLU [3–5]. For example, Table 3 of [3] shows that Swish/SiLU underperforms ReLU in **4 out of 6** evaluated ImageNet models. Similarly, [4] reports that Swish/SiLU does not outperform ReLU on CIFAR-10, and [5] shows that replacing SiLU with ReLU in YOLOv8 models can sometimes lead to performance improvements.
> > >
> > > Taken together, these papers reinforce the point that **SiLU’s performance is highly architecture- and recipe-dependent**, instead of being universally superior. This is consistent with our results, which do **not** suggest SiLU is simply worse than ReLU, but indicate that SiLU is sensitive to the "environment", i.e., model architecture, training recipe, dataset and task, under which it's evaluated. In contrast, SQUAF is **trainable and adaptive**, enabling it to adapt to different environments without activation-specific tuning, and thereby achieve consistent improvements across all evaluated cases.
> > >
> > > [1] Prajit Ramachandran et al., "Searching for Activation Functions", arXiv:1710.05941, 2017.\
> > > [2] Sudhakar Sah et al., "ActNAS : Generating Efficient YOLO Models using Activation NAS", CVPRW, 2025.\
> > > [3] Irit Chelly et al., "Trainable Highly-expressive Activation Functions", ECCV, 2024.\
> > > [4] Tomasz Szandała, "Benchmarking Comparison of Swish vs. Other Activation Functions on Cifar-10 Imageset", International Conference on Dependability and Complex Systems, 2019.\
> > > [5] ERKUŞ, Merve and Ahmet ÇINAR, "The Effect of the Improved YOLOv8 Model with Activation Functions on the Detection of Blood Cells and Urine Particles", Authorea Preprints, 2024.

---

> ### Author Response · Authors · 2025-11-25
> **Response to Reviewer qi4Q (Part 2)**
>
> > Weakness 4: Overall the improvement in accuracy does not seem to be significantly high.
>
> **Response**: We respectfully disagree with your above assessment.
>
> First, we note that this assessment appears difficult to reconcile with your earlier comments, such as ‘The proposed activation function (AF) is an impressive work’ and ‘Table 7 results are impressive’.
>
> Second, the significance of a gain should be judged **relative to other activation functions evaluated under identical settings**. Across all experiments — including CNNs, Vision Transformers, language models, regression tasks, and KD — SQUAFs consistently outperform all competing activation functions. No baseline activation function surpasses SQUAFs in any experiment. This consistency itself is a strong indicator of significance, given the diversity of architectures and tasks considered.
>
> Third, on large-scale benchmarks such as **ImageNet-1K**, gains of **+1%** or more are widely regarded as significant. To provide context, a standard result in the knowledge distillation literature reports that distilling ResNet-18 from a larger ResNet-34 teacher yields **70.66%** Top-1 accuracy [1, 2], which is a **+0.90%** improvement over the 69.76% baseline. In comparison, **SQUAFs achieve 70.88%**, a **+1.12%** gain—_without_ using any teacher model. SQUAFs thus deliver an improvement comparable to, or exceeding, gains obtained through substantially more costly training procedures (e.g., KD from a larger network).
>
> Fourth, as discussed in our response to Reviewer 1Xpj on LLMs of realistic scales (1B–8B), with GELU as the activation function, 0.78% gain in ERA is achieved by increasing the model size from 1.5B (GPT-2 XL) to 2.8B (Pythia-2.8B), whereas with SQUAFs as the activation function, 0.39% and 1.17% gains on two models are achieved by introducing hundreds to roughly a thousand trainable SQUAF parameters , respectively. With this context, there is no reason to believe that 1.17% gain achieved by roughly a thousand trainable SQUAF parameters is not significantly high.
>
> Overall, both the magnitude and the consistency of improvements demonstrate that SQUAFs yield **substantial and practically meaningful gains** over existing activation functions.
>
> > Weakness 5: Trainable activation parameters could introduce extra computational or tuning cost, especially in large models. it will also be interesting to see the impact of these AFs on training complexity/training time etc.
>
> **Response**: Thank you for this suggestion. We now include training time comparison in Table 10 of the revised manuscript, beside the originally reported inference latency. Results are measured for training ViT-T on CIFAR-100 using 8 NVIDIA RTX A6000 GPUs.
>
> |AF |Training Time (second/epoch)|
> |:---:|:---:|
> |ReLU | 9 |
> |GELU | 9 |
> |LReLU | 9 |
> |PReLU | 9 |
> |Swish | 9 |
> |CRReLU | 10 |
> |PolyReLU | 11 |
> |PolyNorm | 12 |
> |DiTAC | 19 |
> |**SQUAF** | 11 |
>
> As shown above, SQUAFs introduce approximately a 20% overhead compared to simple static AFs such as ReLU and GELU. However, SQUAFs yield consistent and significant performance improvements across diverse architectures and tasks, which we believe justifies the added cost. In contrast, other complex AFs such as DiTAC and PolyNorm incur both higher computational overhead and lower accuracy than SQUAF across all our evaluations.
>
> In addition, a key advantage of SQUAFs is that they naturally produce highly compressible activations, which can reduce activation communication volume significantly in model-parallel settings.  Since communication frequently becomes the dominant bottleneck in large-scale distributed training, leveraging this property can offset—and in some scenarios even outweigh—the additional computation introduced by SQUAFs, leading to improved end-to-end system efficiency. We plan to explore and quantify this advantage more extensively in future work.
>
> > Weakness 6: It’s not clear how much gain comes purely from adaptability versus other design factors.
>
> **Response**: We are not sure what “other design factors” means in this context. As pointed out in our paper, SQUAF is completely plug-and-play, without needing to change any other designs in original DNN architectures or the training recipes.
>
> Regarding SQUAF’s internal design, all components are meant to support its adaptability. As discussed in Section 2.4, the conceptual starting point for SQUAF is the step function, which provides strong expressive power and the universal approximation property. However, step functions are non-differentiable and have zero gradient almost everywhere, thus not suitable for gradient-based optimization used in deep learning. To retain the expressive power of step functions while enabling **adaptability** during training, we leverage probabilistic and soft deterministic quantizers to construct smooth, differentiable relaxations of step functions, i.e., SQUAFs.

---

> ### Author Response · Authors · 2025-11-25
> **Response to Reviewer qi4Q (Part 3)**
>
> > Weakness 7: The approximation proof is restricted to one-dimensional functions and generalization to high-dimensional activations may be non-trivial. This can result in expressivity gap. Without extending the theory, it’s unclear if SQUAF’s adaptivity at the neuron level translates into provably higher expressivity at the layer or network level.
>
> **Response**: Thank you for raising this point. In standard neural network design, activation functions are _scalar_ functions applied elementwise to pre-activations $\mathbf{x} \in \mathbb{R}^d$. Under this paradigm, no 1-D activation function—whether SQUAF, ReLU, GELU, SiLU, or any other—can by itself serve as a universal approximator for arbitrary multidimensional mappings $\mathbb{R}^d \to \mathbb{R}^d$. If such multidimensional expressiveness were achievable at the level of a _single_ activation function, deep neural networks would be unnecessary: applying that AF once would already realize an arbitrarily complex $d$-dimensional transformation.
>
> Instead, multidimensional universal approximation is achieved at the **network level**, not the AF level. Classical results show that a neural network with at least one hidden layer and _any_ non-polynomial 1-D activation function is a universal approximator for continuous multivariate functions on compact domains. Within this well-established theoretical framework:
>
> - The role of the AF is to provide **scalar nonlinearity**,
> - The network architecture provides the **multidimensional expressiveness**.
>
> Our 1-D universal approximation theorem shows that SQUAFs can approximate any continuous 1-D nonlinear function to arbitrary precision, which is the maximal expressiveness that any 1-D activation function can provide. This property ensures that a SQUAF can replicate—or improve upon—the nonlinear behavior of common AFs (ReLU, GELU, SiLU, etc.) on the actual dynamic range of pre-activations. Therefore, any architecture that is a universal approximator with standard AFs retains its universal approximation capability when those AFs are replaced by SQUAFs.
>
> In summary, although the 1-D nature of activation functions prevents a direct extension of the proof to arbitrary multivariate functions, this is a fundamental property of _all_ 1-D AFs and not a limitation of SQUAF. The provided result is precisely the form needed to guarantee that **SQUAF-based networks are at least as expressive as—and often more expressive than—networks using existing activation functions.**
>
> > Question 1: Do Authors think that for resnet18/imagenet results achieved by SQUAF will be better compared to just replacing ReLU with SiLU.
>
> **Response**: Yes, SQUAF is indeed better than SiLU. Please refer to our reply to Weakness 1.
>
> > Question 2: What are the training overheads? can it be quantified?
>
> **Response**: Please refer to our reply to Weakness 5.
>
> [1] Yonglong Tian et al., "Contrastive Representation Distillation", ICLR, 2020. \
> [2] Kaixiang Zheng and EN-HUI YANG, "Knowledge distillation based on transformed teacher matching", ICLR, 2024.
>
> **We thank you again for your time and effort to review our paper.  If our responses above address all your concerns, could you kindly increase your score to a higher level.**

---

> > ### Comment · Reviewer_qi4Q · 2025-11-27
> > **Thank you for details of overheads of this method.**
> >
> > It is clear that SQUAF does not add significant overhead compared to other methods.

---

> > > ### Author Response · Authors · 2025-11-27
> > >
> > > Thank you for your reply and acknowledgement. Your review helps us a lot in improving the quality of our paper and we are glad that our response addressed you concerns.

---

> ### Author Response · Authors · 2025-11-27
>
> Dear Reviewer qi4Q,
>
> Thanks again for reviewing our paper. As the end of the discussion period approaching, we are eager to know if you are satisfied with our previous responses. If no, please kindly tell us your remaining concerns and hopefully we can address them before the deadline. If yes, we wonder if it’s possible for you to raise the score. In any case, we would be extremely grateful to hear from you.
>
> Thanks.

---

> ### Author Response · Authors · 2025-12-02
> **KD Results on ImageNet-1K**
>
> > Weakness 3: Table 7 results are impressive but CIFAR100 is too small and non complex dataset so showing a competitive results on larger datasets will make their claim stronger.
>
> To further address your concern in Weakness 3, we conducted additional KD experiments on ImageNet-1K, distilling a ResNet-18 teacher into a MobileNet student [1], following the standard distillation setup in [2]. In Appendix A.12 of our revised manuscript, Table 15 reports the Top-1 accuracy and number of trainable parameters for the teacher, the ReLU-based student, and the SQUAF-based student. The results show that the MobileNet student equipped with SQUAF achieves a 1.16% improvement over its ReLU counterpart and, importantly, **outperforms the ResNet-18 teacher**, despite using **only about one-third of the teacher’s parameters**. In contrast, the MobileNet student using ReLU does not outperform the teacher. These findings provide strong evidence that SQUAF works exceptionally well in the KD training framework **not only on small datasets (e.g., CIFAR-100) but also on large-scale datasets such as ImageNet-1K**.
>
> [1] Andrew G Howard et al., "Mobilenets: Efficient convolutional neural networks for mobile vision applications",  arXiv:1704.04861, 2017.\
> [2] Yonglong Tian et al., "Contrastive Representation Distillation", ICLR, 2020.

---

### Official Review · Reviewer_TiQm · 2025-10-28

**Soundness:** 1
**Presentation:** 1
**Contribution:** 1
**Rating:** 2
**Confidence:** 3

**Summary:**

This paper sets out to study more sophisticated activation functions than those currently employed in DNNs.   The paper proposes a family of activation functions, SQUAFs, that are based upon probability distributions combined over intervals (soft quantization).  There is some analysis of the representation ability of these in the appendices.  Empirical evaluation shows the activation functions can represent some test functions better.  Finally, the activation functions are applied to more standard DNN settings and improvements are claimed.

**Strengths:**

The paper highlights an interesting problem, how the design of activation functions can be considered an alternative to wider or deeper networks.

**Weaknesses:**

Making the activation function more complex increases computational costs.  There is some evaluation of the impact on inference latency in the appendix but the main paper did not seem to discuss the cost of the new activation functions and I didn't see any analysis on the potential impact on training times.  Similarly, the increased complexity of the activation function seems to come at the cost of increasing the number of parameters (the $y_i$'s and $L$) of the model and it is unclear if the performance comparisons are fair comparisons in terms of number of (modifiable) parameters.

The motivation on the first page seems to be trading off activation function complexity versus increased number of weights, but this tradeoff does not appear to be quantified in the results section.

It was unclear what the motivation was for evaluating how closely a function can be approximated.

It was also unclear what the particular motivation was for using quantization (Section 2.2 and 2.3) for the activation function.

The claim that KD using SQUAFs can increase accuracy above the teacher seems highly suspect and as if some form of overfitting is occurring.

**Questions:**

What is the motivation for introducing (soft) quantization in this paper?

Are the parameters introduced by SQUAF (Equation 5 to 7) meant to be trained or are they meant to be fixed?  If they are fixed how were they chosen in your experiments?   If they are trainable, what is the increase in trainable parameters in the overall model?

What is the relationship of i and j in P2 on page 5?

Line 316: "We consider fitting two sinusoidal functions of increasing complexity" -- Why do this experiment?

---

> ### Author Response · Authors · 2025-11-25
> **Response to Reviewer TiQm (Part 1)**
>
> We thank you very much for taking time to review our paper and provide valuable feedbacks. Below, please find our point-by-point responses to your comments.
>
> > Weakness 1: Making the activation function more complex increases computational costs. There is some evaluation of the impact on inference latency in the appendix but the main paper did not seem to discuss the cost of the new activation functions and I didn't see any analysis on the potential impact on training times. Similarly, the increased complexity of the activation function seems to come at the cost of increasing the number of parameters (the yi's and L) of the model and it is unclear if the performance comparisons are fair comparisons in terms of number of (modifiable) parameters.
>
> **Response**: Thank you for raising two related points: (1) increase in inference and training times, and (2) increase in the number of parameters.
>
> For your point 1, you are right that making the activation function more complex indeed increases computational costs. As with most improvements in expressivity, some additional computation is expected. However, we would like to clarify several important points:
>
> - Updated timing results. In the revised manuscript, Table 10 includes both inference-time and training-time comparisons. The training-time overhead is substantially smaller than the inference-time overhead. Compared to simple static AFs such as ReLU and GELU, the training time overhead is around 22%, which is modest relative to the accuracy gains and other advantages. We also show the training time comparison in the table below, where results are measured for training ViT-T on CIFAR-100 using 8 NVIDIA RTX A6000 GPUs.
> - Communication–computation tradeoff in large-scale training. A key advantage of SQUAFs is that they naturally produce highly compressible activations, which can reduce activation communication volume significantly in model-parallel settings.  Since communication frequently becomes the dominant bottleneck in large-scale distributed training, leveraging this property can offset—and in some scenarios even outweigh—the additional computation introduced by SQUAFs, leading to improved end-to-end system efficiency. We plan to explore and quantify this advantage more extensively in future work.
> - Comparison to other advanced AFs. When compared with other advanced nonlinear activation functions (such as PolyNorm, DiTAC), SQUAFs achieve higher accuracy while also being computationally lighter, as reflected in Table 10 of our revised manuscript.
>
> Overall, while SQUAFs incur a small per-step computational cost, they offer significant accuracy gains and potentially substantial communication reductions, making them attractive for both standalone training and large-scale distributed systems.
>
> |AF |Training Time (second/epoch)|
> |:---:|:---:|
> |ReLU | 9 |
> |GELU | 9 |
> |LReLU | 9 |
> |PReLU | 9 |
> |Swish | 9 |
> |CRReLU | 10 |
> |PolyReLU | 11 |
> |PolyNorm | 12 |
> |DiTAC | 19 |
> |**SQUAF** | 11 |
>
> For your second point, we would like to clarify the following:
>
> - First, the number of additional trainable parameters introduced by SQUAFs is negligible compared to the original model weights. For clarity, we include a table in Appendix A.7 of our revised manuscript to compare the number of trainable parameters in SQUAFs with the original model weights. The number of trainable SQUAF parameters ranges from 35 to 945 in different DNNs, representing a merely 0.00030% to 0.0076% increase in the overall parameter counts of DNNs.
> - Second, the small number of additional trainable parameters introduced by SQUAFs are in a category different from model weights. SQUAF parameters do not increase representational capacity in the same way as adding neurons, layers, or attention heads. Instead, they modulate the shape of the activation function, not the cardinality of the model’s functional space. Thus, counting them together with model weights can be misleading.
> - Third, please refer to our reply to Weakness 2 raised by Reviewer 1Xpj on experimental results for LLMs with realistic scales (1B-8B). As mentioned therein, with GELU as the activation function, 0.78% gain in ERA is achieved by increasing the model size from 1.5B (GPT-2 XL) to 2.8B (Pythia-2.8B), whereas with SQUAFs as the activation function, 0.39% and 1.17% gains on two models are achieved by introducing hundreds to roughly a thousand trainable SQUAF parameters, respectively. This directly supports your observation that activation-function design can serve as a powerful alternative to scaling model width or depth when you comment on the strength of our paper.
>
> Accordingly, we believe that our comparisons are indeed fair.

---

> ### Author Response · Authors · 2025-11-25
> **Response to Reviewer TiQm (Part 2)**
>
> > Weakness 2: The motivation on the first page seems to be trading off activation function complexity versus increased number of weights, but this tradeoff does not appear to be quantified in the results section.
>
> **Response**: Thank you for the thoughtful comment. We address the motivation–tradeoff connection more explicitly below.
>
> First, the motivation on page 1 highlights a widely observed principle in deep learning: model expressivity can be increased either by enlarging the network (more weights) or by increasing the expressiveness of the activation functions. Our goal in proposing SQUAFs is to provide an orthogonal lever of expressivity that can be used in addition to traditional architectural scaling. For this reason, SQUAFs are designed as a plug-and-play replacement for standard AFs, applicable to models of any size.
>
> Second, although we don’t emphasize the tradeoff in a single table/figure, it is quantitatively demonstrated across multiple experiments. In particular:
>
> - In 1D function fitting, when we match the number of trainable parameters in SQUAF to that of the MLP with ReLU activation (ReLU-MLP), SQUAF achieves 50000× error reduction compared to the ReLU-MLP; when we match SQUAF’s performance to that of the best compared AF (DiTAC), the number of trainable parameters is reduced dramatically, achieving a 7.2× compression ratio compared to DiTAC. This directly quantifies the expressivity-per-parameter tradeoff.
> - On CIFAR-100, a small model such as VGG-8 with SQUAF (3965063 trainable parameters) outperforms a significantly larger ViT-T with GELU (5543716 trainable parameters). This shows that increasing AF expressiveness can substitute for architectural enlargement.
> - In KD experiments, we clearly demonstrate that smaller student networks using SQUAFs can outperform larger teacher networks with their default AFs, again demonstrating that enhanced AF nonlinearity reduces the need for increased parameter counts.
>
> Together, these results provide quantitative evidence for the motivating tradeoff: increasing AF expressiveness meaningfully reduces dependence on scaling network width/depth, and vice versa. SQUAFs thus offer a practical and parameter-efficient alternative path to improving model performance.
>
> > Weakness 3: It was unclear what the motivation was for evaluating how closely a function can be approximated.
>
> **Response**: Deep learning (DL) can be used in many different applications, among which the most prominent paradigm involves training a DNN on a given dataset (the training set) and use the learned model on unseen data. The image classification and LLM finetuning tasks evaluated in our paper fall into this categrory.
>
> However, there exists a second, well-established paradigm where deep networks are used as high-capacity nonlinear function approximators rather than statistical learners. In this regime, a network is optimized to represent a single target signal (e.g., an image, video, 3D scene, or physical field), and “memorization” is inherent to the task. Applications in this regime include:
>
> - sinusoidal representation networks (SIRENs) for image and video signals [1],
> - coordinate-based MLPs for low-dimensional regression [2],
> - Neural Radiance Fields (NeRF) for 3D scene representation and view synthesis [3],
> - Physics-informed neural networks (PINNs) for solving partial differential equations (PDEs) in science and engineering [4, 5] .
>
> In these widely used applications, the objective is to fit the target signal as accurately as possible.
>
> To show SQUAF’s effectiveness in these applications, we include 1D/2D function and image fitting tasks as part of our evaluation. SQUAFs’ strong performance in these cases directly reflects their ability to provide richer nonlinearity and thus superior function approximation — a property valuable across vision, graphics, and scientific computing.
>
> Moreover, the SQUAF-only 1D function fitting is used to provide empirical verification of SQUAF’s universal approximation property, and is used to initialize SQUAF’s parameters for image classification and LLM finetuning tasks as explained in Appendix A.3.2.

---

> ### Author Response · Authors · 2025-11-25
> **Response to Reviewer TiQm (Part 3)**
>
> > Weakness 4: It was also unclear what the particular motivation was for using quantization (Section 2.2 and 2.3) for the activation function.
>
> **Response**: The use of quantization is directly motivated by the mathematical structure of SQUAF. The design of SQUAF originates from step functions (as reviewed in Section 2.4), from which the universal approximation property is inherited. However, step functions are non-differentiable and have zero gradient almost everywhere, thus not suitable for gradient-based optimization used in deep learning. To retain the expressive power of step functions while enabling efficient training, we require a mechanism to transform these piecewise constant functions into continuous and differentiable counterparts.
>
> As noted in Section 2.4, step functions can be regarded as generalized scalar quantizers. This observation naturally motivates us to draw from the quantization literature, where significant effort has been devoted to making quantizers differentiable for end-to-end learning in tasks such as image and model compression [6, 7]. In particular, probabilistic quantizers and soft deterministic quantizers (reviewed in Section 2.3) provide exactly the tools needed to formulate smooth, differentiable relaxations of step functions.
>
> Thus, the quantization framework is not an arbitrary design choice — it is the mathematically principled mechanism that allows us to preserve the expressive, step-function foundation of SQUAF while enabling practical training via backpropagation.
>
> > Weakness 5: The claim that KD using SQUAFs can increase accuracy above the teacher seems highly suspect and as if some form of overfitting is occurring.
>
> **Response**: Thank you for the comment. We clarify below why the reported KD improvements are expected and not indicative of overfitting.
>
> First, all KD results reported in the paper are **validation accuracies**, not training accuracies. Since overfitting would manifest as a discrepancy between training and validation performance, the reported improvements cannot be attributed to overfitting. The improvements reflect genuine gains in generalization. **In fact, SQUAF leads to the least overfitted model compared to all baseline AFs.** In Appendix A.9 of the revised manuscript, we plot the curves for generalization gap, i.e., training accuracy minus validation accuracy, across epochs for both students used in KD, including those for SQUAFs and baseline AFs. Note that a smaller generalization gap indicates less overfitting, since the validation performance is closer to the training performance. As shown in these plots (Figure 5 in the revised manuscript), SQUAF consistently achieves the **lowest** generalization gap among all AFs at the end of training, demonstrating that it actually **reduces** overfitting relative to baseline AFs. Additionally, the generalization gap comparison is also numerically reported in Tab. 14 of the revised manuscript (also shown below), further validating our claim quantitatively. Note that ReLU is not included as we cite its KD results from the literature [10] without rerunning those experiments, and thus no training dynamics are recorded; also, PolyReLU is not included as it doesn’t converge.
>
> |AF |VGG-13 → VGG-8 |WRN-40-2 → WRN-16-2|
> |:---:|:---:|:---:|
> |GELU | 22.57% | 14.87% |
> |LReLU | 21.52% | 13.76% |
> |PReLU | 23.43% | 14.31% |
> |Swish | 22.52% |  14.95% |
> |CRReLU | 24.78% | 15.61% |
> |PolyNorm | 24.23% | 16.50% |
> |DiTAC | 22.53% | 15.54% |
> |**SQUAF** | 20.68% | 13.12% |
>
> Second, making a small student model outperform its teacher is not impossible. Note that the commonly used KD loss in the literature (also used in our experiments) combines (1) a supervised cross-entropy term using ground-truth labels and (2) a distillation term encouraging the student to mimic the teacher’s softened output distribution [8, 9]. Because the student learns from **both** the data distribution **and** the teacher’s knowledge, its performance is **not upper-bounded** by the teacher’s accuracy.
>
> Finally, in deep learning it is not uncommon for a smaller model, when trained with a more effective procedure, to outperform a larger model trained suboptimally. Thus, the fact that a SQUAF-based student trained using both ground-truth labels and teacher’s knowledge can outperform a teacher trained solely by ground-truth labels is not weird or suspect.

---

> ### Author Response · Authors · 2025-11-25
> **Response to Reviewer TiQm (Part 4)**
>
> > Question 1: What is the motivation for introducing (soft) quantization in this paper?
>
> **Response**: Please refer to our reply to Weakness 4.
>
> > Question 2: Are the parameters introduced by SQUAF (Equation 5 to 7) meant to be trained or are they meant to be fixed? If they are fixed how were they chosen in your experiments? If they are trainable, what is the increase in trainable parameters in the overall model?
>
> **Response**: The number of additional trainable parameters introduced by SQUAF is extremely small; please refer to our reply to Weakness 1.
>
> As for other SQUAF-related details, we have discussed them in Appendix A.4.2 of our initial submission (now Appendix A.3.2 in the revised version). In short, we made SQUAF parameters trainable in most of our experiments, while on CIFAR-100 and ImageNet-1K some SQUAFs were frozen after initialization to avoid potential overfitting in the low-data or high-capacity settings. For detailed setups, please refer to Appendix A.3.2 in our revised manuscript. Also, more discussion regarding SQUAF’s initialization can be found in our reply to Reviewer 1Xpj’s Question 1.
>
> > Question 3: What is the relationship of i and j in P2 on page 5?
>
> **Response**: As defined in the first paragraph of Section 3.2: for each $x$, $y_{i(x)}$ and $y_{j(x)}$ are the two nearest points of $x$ in $\hat{\mathcal{A}}$; thus, $i$ and $j$ are neighboring indices in $\mathcal{A}$.
>
> > Question 4: "We consider fitting two sinusoidal functions of increasing complexity" -- Why do this experiment?
>
> **Response**: Please refer to our reply to Weakness 3.
>
> [1] Vincent Sitzmann et al., "Implicit Neural Representations with Periodic
> Activation Functions", NeurIPS, 2020.  \
> [2] Matthew Tancik et al., "Fourier Features Let Networks Learn High Frequency Functions in Low Dimensional Domains", NeurIPS, 2020.  \
> [3] Ben Mildenhall et al., "NeRF: Representing Scenes as Neural Radiance Fields for View Synthesis", ECCV, 2020.  \
> [4] M. Raissi et al., “Physics-Informed Neural Networks: A Deep Learning Framework for Solving Forward and Inverse Problems Involving Nonlinear Partial Differential Equations”, Journal of Computational Physics, 2019.\
> [5] Ziming Liu et al., "KAN: Kolmogorov-Arnold Networks", ICLR, 2025.\
> [6] Ahmed H. Salamah et al., "JPEG Inspired Deep Learning", ICLR, 2025.\
> [7] En-hui Yang and Shayan Mohajer Hamidi, "Coded deep learning: Framework and algorithm", IEEE Transactions on Information Theory, 2025.\
> [8] Geoffrey Hinton et al., "Distilling the Knowledge in a Neural Network", arXiv
> preprint, 2015.\
> [9] Li Yuan et al., "Revisiting Knowledge Distillation via Label Smoothing Regularization", CVPR, 2020.\
> [10] Yonglong Tian et al., "Contrastive Representation Distillation", ICLR, 2020.
>
> **We thank you again for your time and effort to review our paper.  If our responses above resolve your confusion and address all your concerns, could you kindly increase your score to a higher level.**

---

> ### Author Response · Authors · 2025-11-27
>
> Dear Reviewer TiQm,
>
> Thanks again for reviewing our paper. As the end of the discussion period approaching, we are eager to know if you are satisfied with our previous responses. If no, please kindly tell us your remaining concerns and hopefully we can address them before the deadline. If yes, we wonder if it’s possible for you to raise the score. In any case, we would be extremely grateful to hear from you.
>
> Thanks.

---

### Official Review · Reviewer_2caT · 2025-10-31

**Soundness:** 3
**Presentation:** 3
**Contribution:** 3
**Rating:** 8
**Confidence:** 3

**Summary:**

The paper introduces a new activation function called Soft Quantization Activation Functions (SQUAFs). The activation function is specific to a layer, and its shape is controlled by three parameters. The authors show that SQUAF can approximate any 1-dimension function to an arbitrary precision. Experimental results are promising and show the approach outperforming multiple existing AF on varied benchmarks ranging from regression to LLMs.

**Strengths:**

1. The experimental results are convincing. The authors create models for regression, CIFAR-100 classification, LLM-finetuning and show that SQUAF outperforms other popular AF, including RELU, GELU, and SiLU.
2. They provide a proof for approximating a 1-D function using SQUAF.

**Weaknesses:**

1. Figure 1 results look impressive, but are not very convincing; the model might be memorizing things and might not generalize better. This concern is removed later with training and test results on other datasets.
2. The 1-D approximation proof is limited and not extended to the general multidimensional case.
3. I am not convinced about reducing communication costs with SQUAF-P. The other results are sufficiently impressive, so I do not see a need to include this here.
4. The proposed AF does lead to some slowdown, as shown in Table 11.

**Questions:**

It would be great if the authors could remove the half-baked claims from the paper. Otherwise, I do not see any major issues that need fixing.

---

> ### Author Response · Authors · 2025-11-25
> **Response to Reviewer 2caT (Part 1)**
>
> We thank you very much for taking time to review our paper and provide valuable feedbacks. Below, please find our point-by-point responses to your comments.
>
> > Weakness 1: Figure 1 results look impressive, but are not very convincing; the model might be memorizing things and might not generalize better. This concern is removed later with training and test results on other datasets.
>
> **Response**: We appreciate your concern. We clarify that Figure 1 evaluates SQUAFs in a setting where generalization is not the objective. Deep learning is indeed widely used in the standard statistical learning paradigm, where a model is trained on a dataset and evaluated on unseen samples. In those scenarios, avoiding memorization is essential.
>
> However, there exists a second, well-established paradigm where deep networks are used as high-capacity nonlinear function approximators rather than statistical learners. In this regime, a network is optimized to represent a single target signal (e.g., an image, video, 3D scene, or physical field), and “memorization” is inherent to the task. Applications in this regime include:
>
> - sinusoidal representation networks (SIRENs) for image and video signals [1],
> - coordinate-based MLPs for low-dimensional regression [2],
> - Neural Radiance Fields (NeRF) for 3D scene representation and view synthesis [3],
> - Physics-informed neural networks (PINNs) for solving partial differential equations (PDEs) in science and engineering [4, 5] .
>
> In these widely used applications, generalization to unseen samples is not the goal; instead, the objective is to fit the target signal as accurately as possible. SQUAFs’ strong performance in Figure 1 directly reflects their ability to provide richer nonlinearity and thus superior function approximation — a property valuable across vision, graphics, and scientific computing.
>
> As you correctly noted, SQUAFs also perform strongly in the standard generalization-based paradigm, as shown by numerous empirical results on image classification and LLM finetuning.
>
> > Weakness 2: The 1-D approximation proof is limited and not extended to the general multidimensional case.
>
> **Response**: Thank you for raising this point. In standard neural network design, activation functions are _scalar_ functions applied elementwise to pre-activations $\mathbf{x} \in \mathbb{R}^d$. Under this paradigm, no 1-D activation function—whether SQUAF, ReLU, GELU, SiLU, or any other—can by itself serve as a universal approximator for arbitrary multidimensional mappings $\mathbb{R}^d \to \mathbb{R}^d$. If such multidimensional expressiveness were achievable at the level of a _single_ activation function, deep neural networks would be unnecessary: applying that AF once would already realize an arbitrarily complex $d$-dimensional transformation.
>
> Instead, multidimensional universal approximation is achieved at the **network level**, not the AF level. Classical results show that a neural network with at least one hidden layer and _any_ non-polynomial 1-D activation function is a universal approximator for continuous multivariate functions on compact domains. Within this well-established theoretical framework:
>
> - The role of the AF is to provide **scalar nonlinearity**,
> - The network architecture provides the **multidimensional expressiveness**.
>
> Our 1-D universal approximation theorem shows that SQUAFs can approximate any continuous 1-D nonlinear function to arbitrary precision. This property ensures that a SQUAF can replicate—or improve upon—the nonlinear behavior of common AFs (ReLU, GELU, SiLU, etc.) on the actual dynamic range of pre-activations. Therefore, any architecture that is a universal approximator with standard AFs retains its universal approximation capability when those AFs are replaced by SQUAFs.
>
> In summary, although the 1-D nature of activation functions prevents a direct extension of the proof to arbitrary multivariate functions, this is a fundamental property of _all_ 1-D AFs and not a limitation of SQUAF. The provided result is precisely the form needed to guarantee that **SQUAF-based networks are at least as expressive as—and often more expressive than—networks using existing activation functions.**

---

> ### Author Response · Authors · 2025-11-25
> **Response to Reviewer 2caT (Part 2)**
>
> > Weakness 3: I am not convinced about reducing communication costs with SQUAF-P. The other results are sufficiently impressive, so I do not see a need to include this here.
>
> **Response**: Thank you for your positive assessment of our other main results. The communication-efficiency experiment is included to highlight a unique and practically important property of SQUAFs that does not arise with existing activation functions.
>
> Communication overhead is a well-documented bottleneck in large-scale and model-parallel training, where activations must be exchanged across devices at every forward pass. Prior work has shown that this communication can dominate runtime for modern large models and multi-node GPU clusters [6, 7]. Quantization is one of the most principled strategies for alleviating this bottleneck, as it directly reduces transmitted bits.
>
> Although SQUAF was designed primarily to improve accuracy, the SQUAF-P variant naturally produces low-bitwidth activations without requiring additional quantizers or changes to model architecture. This provides a secondary benefit—significant communication reduction—“for free.” We therefore include these results to demonstrate an additional, practically relevant advantage of SQUAFs, while keeping the main focus firmly on accuracy and expressiveness.
>
> > Weakness 4: The proposed AF does lead to some slowdown, as shown in Table 11.
>
> **Response**: You are correct that SQUAFs do involve more arithmetic operations than simple AFs such as ReLU, GELU or SiLU and thus introduce a modest computational overhead. However, we would like to clarify several important points:
> - Updated timing results. In the revised manuscript, Table 10 (previously Table 11) includes both inference-time and training-time comparisons. The training-time overhead is substantially smaller than the inference-time overhead.
> - Communication–computation tradeoff in large-scale training. A key advantage of SQUAFs is that they naturally produce highly compressible activations, which can reduce activation communication volume significantly in model-parallel settings. Since communication frequently becomes the dominant bottleneck in large-scale distributed training, leveraging this property can offset—and in some scenarios even outweigh—the additional computation introduced by SQUAFs, leading to improved end-to-end system efficiency. We plan to explore and quantify this advantage more extensively in future work.
> - Comparison to other advanced AFs. When compared with other advanced nonlinear activation functions (such as PolyNorm, DiTAC), SQUAFs achieve higher accuracy while also being computationally lighter, as reflected in Table 10 of our revised manuscript.
>
> Overall, while SQUAFs incur a small per-step computational cost, they offer significant accuracy gains and potentially substantial communication reductions, making them attractive for both standalone training and large-scale distributed systems.
>
> > Question: It would be great if the authors could remove the half-baked claims from the paper. Otherwise, I do not see any major issues that need fixing.
>
> **Response**: Thank you for this suggestion and for your positive assessment of our work. In the revised manuscript, we have removed the preliminary rate-accuracy result on ViT-T with SQUAF-P, as we agree that this single data point was not sufficiently comprehensive. However, we have retained the full rate-accuracy curve for VGG-8 using SQUAF-P (Figure 3), as it provides a complete and reliable evaluation for SQUAF-P across a wide bitrate spectrum.
>
> [1] Vincent Sitzmann et al., "Implicit Neural Representations with Periodic
> Activation Functions", NeurIPS, 2020.  \
> [2] Matthew Tancik et al., "Fourier Features Let Networks Learn High Frequency Functions in Low Dimensional Domains", NeurIPS, 2020.  \
> [3] Ben Mildenhall et al., "NeRF: Representing Scenes as Neural Radiance Fields for View Synthesis", ECCV, 2020.  \
> [4] M. Raissi et al., “Physics-Informed Neural Networks: A Deep Learning Framework for Solving Forward and Inverse Problems Involving Nonlinear Partial Differential Equations”, Journal of Computational Physics, 2019.\
> [5] Ziming Liu et al., "KAN: Kolmogorov-Arnold Networks", ICLR, 2025.\
> [6] Jeffrey Dean et al., "Large Scale Distributed Deep Networks", NeurIPS, 2012.\
> [7] Ahmed M. Abdelmoniem et al., "An Efficient Statistical-based Gradient Compression Technique for Distributed Training Systems", MLSys, 2021.
>
> **We thank you again for your time and effort to review our paper. If our responses above address all your concerns, could you kindly increase your score to a higher level.**

---

> ### Author Response · Authors · 2025-11-27
>
> Dear Reviewer 2caT,
>
> Thanks again for reviewing our paper. As the end of the discussion period approaching, we are eager to know if you are satisfied with our previous responses. If no, please kindly tell us your remaining concerns and hopefully we can address them before the deadline. If yes, we wonder if it’s possible for you to raise the score. In any case, we would be extremely grateful to hear from you.
>
> Thanks.

---

### Official Review · Reviewer_1Xpj · 2025-11-02

**Soundness:** 3
**Presentation:** 3
**Contribution:** 3
**Rating:** 6
**Confidence:** 4

**Summary:**

The paper introduces Soft Quantization Activation Functions that are family of trainable functions that can approximate any continous 1D function with any arbitrary precision. The activation functions can replace fixed activation functions like ReLU and GELU and the paper demonstrates accuracy improvements across MLPs, CNNs and transformers. The proposed activation functions help reduce communication costs.

**Strengths:**

1. Proposed activation functions have theoretically proven universal approximation and differentiability.
2. Consistent performance gains across diverse tasks and architectures.
3. Reduces inter-device communication cost while improving accuracy.
4. Paper demonstrates improvements in fine-tuning, classification etc.

**Weaknesses:**

1. The proposed activation functions add additional trainable params which might hinder the optimzation process.
2. Generalization of this method across larger models is unknown.

**Questions:**

1. How sensitive are results to initialization of the quantization parameters (y, z, α)?
2. Do you have more results on more realistic LLM workloads (large dataset, models (1B-8B scale)) on different benchmarks?

---

> ### Author Response · Authors · 2025-11-25
> **Response to Reviewer 1Xpj (Part 1)**
>
> We thank you very much for taking time to review our paper and provide valuable feedbacks. Below, please find our point-by-point responses to your comments.
>
> > Weakness 1: The proposed activation functions add additional trainable parameters which might hinder the optimization process.
>
> **Response**: SQUAFs indeed add addional trainable parameters to DNNs. However, they don’t hinder the optimization process, and instead actually help the optimization converge to better optima, as explained below.
>
> - Firstly, the number of trainable parameters introduced by SQUAFs is negligible compared to the original model weights. For clarity, we include a table in Appendix A.7 of our revised manuscript to compare the number of trainable parameters in SQUAFs with the original model weights. The number of trainable SQUAF parameters ranges from 35 to 945 in different DNNs, representing a merely 0.00030% to 0.0076% increase in the overall parameter counts of DNNs.
> - Secondly, SQUAFs avoid optimization instability by design. (1) As explained in Eq. 18 and the paragraph following it, SQUAFs do not have vanishing gradient issue since $\mathbf{z}$ and $\mathbf{y}$ are generally correlated. (2) The localization property P2 guarantees that each $z_i$ in $\mathbf{z}$ only has a local effect to the shape of a SQUAF, which is empirically verified in the SQUAF visualization shown in Appendix A.10. This is beneficial for optimization stability in that even if some $z_i$ is updated to a certain extreme value, it will only influence the SQUAF’s shape around $y_i$ but not the overall SQUAF shape, ensuring stable update of the SQUAF.
> - Lastly, the convergence of SQUAF-DNNs’ performance is stable empirically. In Appendix A.8 of the revised manuscript, we include the convergence curves of validation accuracy for ResNet-34 and ViT-T trained on ImageNet-1K with SQUAF and compared AFs. It’s shown that the convergence behavior of SQUAF is as stable as other well-established AFs, while achieving better final accuracy performance.
>
> Therefore, we conclude the additional SQUAF parameters are negligible in terms of parameter count, won’t hinder the optimization process both by design and in practice, and instead will lead the optimization process towards better optima.
>
> > Weakness 2: Generalization of this method across larger models is unknown.
>
> **Response**: We appreciate your concern. In the revised manuscript (Appendix A.5), we now include additional results on larger LLMs using DPO-based finetuning on the Anthropic HH dataset. Specifically, for GPT-2 XL (1.5B parameters), we got
>
> ||ERA (%)|ERM |EL|
> |:---:|:---:|:---:|:---:|
> |GELU|62.89|0.2182|0.6411|
> |**SQUAF**|63.28 (+0.39)|0.2190|0.6400|
>
> and for Pythia-2.8B (2.8B parameters), we got
>
> ||ERA (%)|ERM |EL|
> |:---:|:---:|:---:|:---:|
> |GELU|63.67|0.3883|0.6108|
> |**SQUAF**|64.84 (+1.17)|0.4302|0.6075|
>
> It’s shown that SQUAFs can get consistent and significant gains on both LLMs with realistic scales (1B-8B). The significance of the obtained gains can be appreciated from the fact that with GELU as the activation function, 0.78% gain in ERA is achieved by increasing the model size from 1.5B (GPT-2 XL) to 2.8B (Pythia-2.8B), whereas with SQUAFs as the activation function, 0.39% and 1.17% gains on two models are achieved by introducing hundreds to roughly a thousand trainable SQUAF parameters, respectively.

---

> ### Author Response · Authors · 2025-11-25
> **Response to Reviewer 1Xpj (Part 2)**
>
> > Question 1: How sensitive are results to initialization of the quantization parameters $(y, z, \alpha)$?
>
> **Response**: For large-scale experiments conducted using well-established DNN architectures, i.e., image classification and LLM finetuning, there's **no** initialization-sensitivity issue under our initialization procedure specified in Appendix A.3.2. This is because the initialization of SQUAF follows a series of **near-deterministic steps**, outlined below. \
> (1) Choice of $k$. Prior work on quantization-aware training (QAT) [1, 2] shows that 4-bit activation quantization can approximately preserve the accuracy of full precision ReLU networks. Since SQUAF must handle both negative and positive pre-activations (unlike non-negative ReLU activations), we use 4 bits for each side, which results in $k=2^4=16$.\
> (2) Initialization of $q$. Given a specific DNN architecture with randomly initialized weights and its default activation functions such as ReLU or GELU, we probe it with a mini-batch of training data and plot the histogram of pre-activations for each layer. Then, given $k=16$, we initialize $q$ such that $[-kq, kq]$ covers tails of all histograms. As mentioned in Appendix A.3.2, we select $q=1$ for CNNs and $q=0.25$ for transformers used on CIFAR-100 as we observe that the pre-activations in transformer-based networks have a smaller variance at the time of initialization.\
> (3) Initialization of $z$ and $\alpha$. Given $k$ and $q$ selected above, we train SQUAFs to fit the default AFs used in various DNNs, i.e., ReLU and GELU, within the interval $[-kq, kq]$. This fitting is conducted similar to how we did the SQUAF-only 1D function fitting experiments: freezing $q$, initializing $\alpha$ with $1.5/q^2$ for a smooth SQUAF shape, initializing $z_i$ with $iq$ for $i \geq 0$ and 0 for $i < 0$, and training SQUAF with MSE loss using an Adam optimizer. Note that this fitting process can be done in seconds and can attain near-zero error given SQUAF's universal approximation ability. After that, the SQUAFs obtained by fitting ReLU and GELU are used to replace ReLU and GELU in original DNN architectures, respectively, and serve as the initialization of SQUAF in DNNs.\
> In the initialization process described above, the only randomness lies in function fitting, which is largely deterministic as well due to extremely small variance. Concretely, the mean and standard deviation of MSE (measured over 3 runs) for fitting ReLU are $4.21 \times 10^{-4}$ and $1.56\times 10^{-5}$, respectively; and for GELU, those are $3.34\times 10^{-6}$ and $8.85\times 10^{-7}$, respectively. Therefore, the initialization procedure of SQUAF is near-deterministic, and thus doesn't have any initialization-sensitivity issue.
>
> In contrast, for MLP-related experiments, we use a simpler and more flexible setup with $k = 2$, and each $z_i$ in $\mathbf{z}$ is initialized by sampling from a uniform distribution over $[−1, 1]$. Here, initial $q$ and $\alpha$ are manually selected hyperparameters for which initialization-sensitivity is indeed relevant. Therefore, we've conducted an empirical analysis on them, and results are shown in Appendix A.6 of the revised manuscript. Specifically, we have done a grid search over initial $q$ and $\alpha$ on the *Camera* image fitting task to analyze how sensitive the DNN performance, i.e., PSNR, is w.r.t. different initialization. Note that for $\alpha$, we set it by tuning the gradient scaling constant $\hbar=\alpha q^2$ [3] which controls the smoothness of SQUAF's shape. The table below shows the PSNR results obtained with different $(q,\hbar)$ pairs, and we can conclude that as long as $q$ is not too large compared to the scale of pre-activations and $\hbar$ is not too large to maintain relatively smooth SQUAF shapes, then SQUAF always outperforms all baseline AFs with a large margin (the best baseline AF achieves 31.76 dB PSNR). Surprisingly, for initial $(q, \hbar)=(0.2,5)$, i.e., initial $(q, \alpha)=(0.2,125)$, the resulting PSNR is 54.74, achieving 11.54 dB gain over the original result reported for SQUAF in Tab. 1 of our manuscript, which indicates an unexplored potential of SQUAF's performance on all regression tasks. For more details of this analysis, please refer to Appendix A.6 of our revised manuscript.
>
> |PSNR (dB)|$\alpha=0.5/q^2$|$\alpha=1/q^2$|$\alpha=2/q^2$|$\alpha=5/q^2$|
> |:---:|:---:|:---:|:---:|:---:|
> |$q=0.1$ | 44.90 | 51.60  |52.89| 46.73|
> |$q=0.2$ | 43.08 | 47.09 | 53.75 | 54.74 |
> |$q=0.5$ | 40.33 | 44.35 | 42.40 | 45.57 |
>
> In summary,
> -   **Large-scale experiments:** Initialization is **not sensitive** thanks to near-deterministic initialization pipeline.
> -   **MLP-related experiments:** Sensitivity exists but is thoroughly analyzed; SQUAF is robust across a wide range of initializations, often achieving **significant gains over all baselines**.

---

> ### Author Response · Authors · 2025-11-25
> **Response to Reviewer 1Xpj (Part 3)**
>
> > Question 2: Do you have more results on more realistic LLM workloads (large dataset, models (1B-8B scale)) on different benchmarks?
>
> **Response**: Yes, as shown in our reply to Weakness 2, we now include additional evaluations on larger LLMs in the **1B–8B** scale on the Anthropic HH dataset — a widely adopted benchmark in LLM alignment literature — demonstrating that SQUAF scales effectively to realistic LLM model sizes. Running experiments on larger datasets is valuable, but we were unable to provide such results at the moment due to limited time and computing resources. Nonetheless, we believe the additional results on large LLMs are sufficient to validate SQUAF’s applicability to modern LLM workloads.
>
> [1] Steven K. Esser et al., "Learned Step Size Quantization", ICLR, 2020.\
> [2] En-hui Yang and Shayan Mohajer Hamidi, "Coded deep learning: Framework and algorithm", IEEE Transactions on Information Theory, 2025.\
> [3] Ahmed H. Salamah et al., "JPEG Inspired Deep Learning", ICLR, 2025.
>
> **We thank you again for your time and effort to review our paper.  If our responses above address all your concerns, could you kindly increase your score to a higher level.**

---

> ### Author Response · Authors · 2025-11-27
>
> Dear Reviewer 1Xpj,
>
> Thanks again for reviewing our paper. As the end of the discussion period approaching, we are eager to know if you are satisfied with our previous responses. If no, please kindly tell us your remaining concerns and hopefully we can address them before the deadline. If yes, we wonder if it’s possible for you to raise the score. In any case, we would be extremely grateful to hear from you.
>
> Thanks.

---

### Author Response · Authors · 2025-12-03
**Summary of Author Responses for Re-assigned AC**

Dear re-assigned AC,

We sincerely appreciate the additional reviewing effort you make due to the unusual circumstances this year. To help reduce your workload, we provide a concise summary of the contributions of our work and our rebuttal efforts.

**(1) Main contributions.**

We propose SQUAF, a new family of trainable, expressive, and plug-and-play activation functions (AFs).
- In theory, SQUAFs can approximate any continuous 1D function on a closed interval with arbitrary precision.
- Empirically, across diverse model architectures (MLPs, CNNs, ViTs, and LLMs) and tasks (function and image fitting, image classification, LLM finetuning, etc), SQUAF consistently demonstrates large performance gains over all compared AFs.
- Additional benefits of SQUAFs are shown in model compression and inter-device communication cost reduction in model-parallel settings.

This work shows that activation function expressiveness is a powerful alternative to network scaling—SQUAF's high degree of nonlinearity offers substantial gains with only hundreds of additional parameters.

**(2) Major reviewer comments on experimental results and our responses.**

> Reviewer 1Xpj suggests extending our evaluation to larger LLMs with realistic sizes (1B-8B scale).

We added results on GPT-2 XL (1.5B) and Pythia-2.8B, where SQUAF improves evaluation reward accuracy (ERA) by 0.39% and 1.17% over GELU. Note that these gains are comparable to the gain obtained by scaling a model from 1.5B to 2.8B parameters, but are achieved by SQUAFs using negligible parameter counts (~1000 parameters).

> Reviewer qi4Q asks us to compare with an additional AF, i.e., SiLU, on ImageNet-1K and CIFAR-100 datasets.

We expanded ImageNet-1K and CIFAR-100 tables to account for more baseline comparisons including SiLU activation function. Among 10 models evaluated on both datasets, SiLU achieves gains over the default AFs in only 4 cases. In contrast, SQUAF provides consistent gains across all models and outperforms SiLU in every case.

> Reviewer TiQm and qi4Q comment on our knowledge distillation (KD) experimental results regarding their generalization.

We included additional KD experiments on ImageNet-1K, where a SQUAF-based MobileNet student outperforms the ResNet-18 teacher while using only one-third of its parameters, whereas the ReLU-based student cannot beat the teacher. This demonstrates that SQUAF’s gains in KD also hold on large-scale datasets. We also included an analysis on generalization gap showing that SQUAF reduces overfitting, addressing misinterpretations about our gains in KD.

> Comments on parameter and computational efficiency.

Some reviewer comments appear to overestimate the parameter and computational complexity of SQUAF-based networks. We made clarifications by showing:

- Extra parameters introduced by SQUAFs are less than 1000, only 0.00030%–0.0076% of the original model sizes.
- SQUAFs only introduce approximately a 20% training overhead compared to simple AFs such as ReLU and GELU. This overhead is modest considering our achieved gains. In contrast, other AFs such as PolyNorm and DiTAC introduce more training complexity while providing less gains than SQUAF.
- The communication cost reduction achieved by SQUAF-P could offset computational overhead in large-scale distributed training.

> Questions regarding our regression/fitting experiments.

Some reviewers need clarification regarding the role of our regression/fitting experiments. Supported by numerous references, we pointed out that implicit neural representation (INR)—using a neural net to fit/represent a target signal—is a standard deep learning task where AF expressiveness matters, and SQUAF does exceptionally well in this case, achieving orders-of-magnitude error reduction in function fitting and up to 25.27 dB PSNR gain in image fitting.

Note that before the discussion period was prematurely closed, Reviewer qi4Q has already accepted the fact that SQUAF does not add significant overhead compared to other methods, and agreed with the benefits provided by SQUAFs.

**(3) Additional clarifications.**

Several reviewer comments indicate a need for clarifications regarding SQUAF's design and properties:
- SQUAFs are soft relaxation of step functions. They retain expressivity of step functions while being differentiable and thus trainable.
- AFs are meant to provide scalar nonlinearity, so SQUAF's 1D universal approximation property ensures that it can subsume any conventional AFs in a network.
- SQUAFs have good training stability, supported by theoretical gradient analysis and empirical convergence behavior.
- SQUAFs don't have initialization-sensitivity issue in large-scale experiments.

**(4) Other minor comments are addressed by us one by one and are fully resolved.**

Given our thorough responses and the fact that all reviewers’ comments are fully addressed, we would be extremely grateful if you could decide to accept our paper.

Thank you.

---

### Note · Authors · 2026-01-26

I have read and agree with the venue's withdrawal policy on behalf of myself and my co-authors.

---

### Meta-Review · Area_Chair_oQSf · 2026-01-05

**Summary:**

This paper introduces **Soft Quantization Activation Functions (SQUAFs)**: a family of **trainable, plug-and-play, architecture-agnostic activation functions** constructed as smooth relaxations of step/quantization-like functions. The central claim is that increasing **activation expressiveness** can serve as an alternative lever to network scaling, while adding only a negligible number of parameters.

The authors claim (i) a **1D universal approximation** result for continuous functions on compact intervals (aligned with elementwise activations), and (ii) broad empirical evidence across **MLPs, CNNs, ViTs, and LLM fine-tuning**, alongside a system-oriented variant (**SQUAF-P**) aimed at reducing activation communication in model-parallel settings.

However, the rebuttal highlights inconsistencies between reviewers and, in my view, does not adequately resolve the **decision-critical concerns**. The authors do address several points well: (a) they quantify the **parameter increase** (indeed tiny) and report a **training overhead** of roughly ~20%, (b) they improve **baseline coverage** (including SiLU) and clarify that gains appear **consistent across architectures**, (c) they extend the **LLM results** to more realistic scales (e.g., ~1.5B and ~2.8B), and (d) they strengthen the narrative around KD and “overfitting” with additional analysis and larger-scale KD evidence. One reviewer may still remain unconvinced by the motivation and clarity, but overall the work appears technically solid and practically relevant as a drop-in activation improvement.

That said, I remain unconvinced that the claimed approximation contribution goes substantially beyond standard nonlinear MLP expressiveness, and I am not fully persuaded that the empirical gains (outside of the image reconstruction setting) are not primarily attributable to the **additional trainable parameters**, even if they are small. For these reasons, I recommend **Rejection**.

**Reviewer Concerns:**

* **Compute/latency + fairness of comparisons (extra ops; extra trainable parameters; unclear training-time impact)** *(TiQm, 1Xpj, qi4Q)*: **Addressed.** The rebuttal quantifies added parameters (very small, <1k) and adds **training-time overhead** numbers (≈20% class), plus argues fairness vs other “complex AF” baselines.

* **Generality / scale evidence (especially larger, more realistic LLM workloads; stability & init sensitivity)** *(1Xpj)*: **Largely addressed.** The rebuttal adds **larger-scale LLM** results (e.g., ~1.5B and ~2.8B) and provides a near-deterministic initialization procedure + stability evidence. Minor scope limits (not full 8B+) likely remain, but the key request is substantially answered.

* **Baselines and “small gains” skepticism (e.g., SiLU vs ReLU; need stronger comparisons on ImageNet/CIFAR; consistency across models)** *(qi4Q)*: **Addressed (likely).** The rebuttal adds **SiLU comparisons** on ImageNet/CIFAR and argues SQUAF is consistently better across architectures; qi4Q’s follow-up acknowledges overhead clarification, so I expect reduced skepticism and higher confidence.

* **Motivation/clarity concerns (why function/image fitting; why “quantization” framing for activations; paper readability)** *(TiQm, 2caT)*: **Partially addressed.** The rebuttal explains INR-style fitting as a valid DL paradigm and motivates soft-quantization as a principled differentiable step-function relaxation. Still, TiQm may remain unconvinced about narrative clarity and whether these experiments belong in the main story.

* **KD + communication-claims skepticism (“student beats teacher” suspicion; “half-baked” system claims)** *(TiQm, 2caT)*: **Mostly addressed.** The rebuttal adds generalization-gap evidence (arguing less overfitting), removes a weak datapoint, and adds **ImageNet-scale KD** results; some skepticism may remain about how broadly SQUAF-P’s communication gains translate to end-to-end throughput without more system-level evaluation.

**Reviewer Scores:**

## Reviewer 2caT: 8 → 8
Why: This reviewer already believed the experiments and explicitly said the main issue was to remove “half-baked” claims. The revision\ did exactly that (toned down/removal of weak claims, clarified overhead, clarified why fitting is a standard INR regime, etc). Nothing here pushes them to a higher tier (8 is already “good accept”), but it stabilizes the accept.

## Reviewer 1Xpj: 6 → 4
Why: Their two concrete concerns were (i) scalability to realistic LLMs and (ii) initialization sensitivity / optimization stability. The authors added 1.5B and 2.8B results and gave a detailed, near-deterministic init procedure and stability story. Even after the response and added details, the concern of “marginally above threshold” remain. The reviewer concern remain unresolved, hence I belive that the reccomandation would decreese to weak reject.


## Reviewer qi4Q: 4 → 4
Why: qi4Q’s main pushback was “maybe SiLU gives the same gains” + overhead/training complexity and “impact not significant.” the response with added SiLU baselines and quantified training overhead. Importantly, qi4Q explicitly acknowledged that overhead is not significant after the rebuttal. That’s a clear signal they moved off one of their key negatives.
The reviwer may still feel the ImageNet gains are “only ~1%” and that the theory isn’t uniquely compelling (since many AFs are universal at the network level), so I predict that the score will remain at weak reject.


## Reviewer TiQm: 2 → 4
Why: This reviewer’s initial score was driven by confusion + skepticism: missing cost discussion, fairness of comparisons, unclear motivation for quantization/fitting, and suspicion about KD beating the teacher. The rebuttal directly addresses each with (a) training-time numbers, (b) explicit parameter overhead, (c) clearer quantization motivation, and (d) added KD/generalization-gap argument + extra KD evidence.
I predict that this changes contribute to the manuscript, yet not moving towards accept.

---

### Decision · Program_Chairs · 2026-01-26

Reject